

# Grain-size evolution controls the accumulation dependence of modeled firn thickness

Jonathan Kingslake, Robert Skarbek, Elizabeth Case, and Christine McCarthy

Lamont-Doherty Earth Observatory, Department of Earth and Environmental Science, Columbia University, New York, NY, 10027

**Correspondence:** J. Kingslake (j.kingslake@columbia.edu)

**Abstract.** The net rate of snow accumulation $b$ is predicted to increase over large areas of the Antarctica and Greenland ice sheets as the climate warms. Models disagree on how this will affect the thickness of the firn layer – the relatively low-density upper layer of the ice sheets that influences altimetric observations of ice-sheet mass change and paleo-climate reconstructions from ice cores. Here we examine how $b$ influences firn compaction and porosity in a simplified model that accounts for

mass conservation, dry firn compaction, grain size evolution, and the impact of grain size on firn compaction. Treating $b$ as a boundary condition and employing an Eulerian reference frame helps to untangle the factors controlling the $b$-dependence of firn thickness. We present numerical simulations using the model as well as simplified steady-state approximations to the full model, to demonstrate how the downward advection of porosity and of grain size are both affected by $b$, but have opposing impacts on firn thickness. The net result is that firn thickness increases with $b$ and that the strength of this dependence increases

with the surface grain size. We also quantify the circumstances under which porosity- and grain-size-advection balance exactly, which counter-intuitively renders steady-state firn thickness independent of $b$. These findings are qualitatively independent of the stress-dependence of firn compaction and whether the thickness of the ice-sheet is increasing, decreasing, or steady. They do depend on the grain-size dependence of firn compaction. Firn models usually ignore grain-size evolution, but we highlight the complex effect it can have on firn thickness when included in a simplified model. This work motivates future efforts to

better observationally constrain the rheological effect of grain size in firn.

## 1 Introduction

Firn is snow that has persisted for at least one full year on the surface of a glacier or ice sheet. In the absence of significant surface melting, firn is transformed into glacial ice through dry firn compaction. As it is buried by subsequent snow fall, the vertical load of the overlying material compacts firn until it becomes glacial ice (e.g., Cuffey and Paterson, 2010). Understand-

ing firn compaction is important for dating gases trapped in ice cores (e.g., Schwander and Stauffer, 1984; Parrenin et al., 2012; Buizert et al., 2015), reconstructing past temperatures from ice core records (e.g., Buizert et al., 2021), and estimating present-day ice sheet mass change (e.g., Helsen et al., 2008; Smith et al., 2020).

     Particularly important is understanding how the thickness of the firn layer will respond to changes in temperature and the rate of snow accumulation (Herron and Langway, 1980; Helsen et al., 2008; Buizert et al., 2021). Both surface forcings are





predicted to increase as the climate warms (Frieler et al., 2015; Kittel et al., 2021), but models of firn compaction disagree on
how this will affect firn thickness. Firn thickness can be characterized by the distance from the surface to where firn reaches
a density of $830 \, \mathrm{kg \, m^{-3}}$, $z_{830}$, approximately where gas bubbles become isolated from one another (van den Broeke, 2008).
A competition between compaction rate and downward advection of low-density surface firn controls $z_{830}$, with increased
downward advection increasing $z_{830}$ and increased compaction rate decreasing $z_{830}$. Compaction rate increases with the surface
temperature $T_s$ because the micro-processes that facilitate compaction are more efficient at higher temperatures (Herron and
Langway, 1980). While disagreement regarding the strength of this relationship and the most appropriate way to describe it
mathematically remains (e.g., Zwally and Jun, 2002; Li and Zwally, 2015), there is widespread agreement that higher $T_s$ leads
to a smaller $z_{830}$.

Less consensus exists regarding the dependence of $z_{830}$ on the net rate of snow accumulation on the ice-sheet surface $b$.
Higher $b$ speeds up downward advection of low-density surface firn, which thickens the firn layer. However, confusion sur-
rounds the impact of $b$ on compaction rates. For example, Zwally and Jun (2002) stated that the rate of increase of overburden
stress on a parcel of firn increases with $b$, so increasing $b$ accelerates compaction, decreasing $z_{830}$. However, if firn compaction
is viscous it is the overburden stress, not the rate of increase in overburden stress that drives compaction. These contrasting
perspectives are reflected in the differing formulation of firn compaction models. These models employ constitutive relations
describing compactive strain rates (vertical deformation due purely to compaction, rather than horizontal ice-sheet flow, for
example; Horlings et al. (2021)) to simulate firn densities given prescribed environmental conditions, including surface temper-
ature, accumulation, or surface grain size. Unfortunately for attempts to untangle the impact of $b$ on $z_{830}$, models fundamentally
differ in how they include $b$. Some (Groot Zwaaftink et al., 2013; Arnaud et al., 2000; Arthern and Wingham, 1998) treat ac-
cumulation as a boundary condition, as it is in other ice-deformation modelling contexts; Schoof and Hewitt (2013). Others
(Zwally and Jun, 2002; Helsen et al., 2008; Li and Zwally, 2004, 2015; Medley et al., 2020) include $b$ in their constitutive
relations. This was first motivated by Herron and Langway (1980). Their Equation 4a describes compaction rate as follows:

$$\frac{D\rho}{Dt} = Cb^{\alpha}(\rho_i - \rho), \tag{1}$$

where $\rho$ is firn density, $\rho_i$ is ice density, $t$ is time, $D/Dt$ is the material derivative, $C$ is a constant that depends on tem-
perature and $\alpha$ is a constant that Herron and Langway (1980) found to be approximately one in their low-density regime
($\rho < 550 \, \mathrm{kg \, m^{-3}}$). A limitation of including $b$ in the constitutive relation is that it causes compaction rates to respond instan-
taneously throughout the firn column to changes in $b$. This is unrealistic when $b$ varies on timescales similar to or shorter than
the time taken for the firn layer to reach a steady state (Li and Zwally, 2015; Stevens et al., 2020).

Starting from a full dynamic model of firn compaction including grain-size advection and growth, Arthern et al. (2010) (in
their Appendix B) provided physical justification for Eq. 1 by assuming a steady state and a negligibly small grain size at the
surface. The implication is that models that employ a formulation based on Eq. 1 implicitly make assumptions about grain size
and its evolution that have not been examined in detail. Moreover, the inclusion of $b$ in many models' constitutive relations,
combined with the fact that most take a Lagrangian approach, which tracks each firn layer individually, obscuring the role of
advection, makes unravelling the influence of $b$ and grain size on $z_{830}$ using such models challenging.





**Table 1.** Model variables and coordinates.

| variable | description | units |
|---|---|---|
| $A$ | age | s |
| $b$ | accumulation rate | m s$^{-1}$ |
| $h$ | domain height | m |
| $\sigma$ | overburden stress | Pa |
| $\phi$ | bulk porosity | - |
| $t$ | time | s |
| $T_s$ | surface temperature | K |
| $r$ | grain radius | m |
| $\rho$ | density | kg m$^{-3}$ |
| $w$ | vertical velocity | m s$^{-1}$ |
| $z$ | depth | m |

In this paper we aim to explore the implications of the assumptions described above, and elucidate how firn thickness depends
on accumulation and grain size in simple firn compaction models. We present and analyze an Eulerian firn compaction model
based on Arthern et al. (2010) and Case and Kingslake (2021), along with reduced, steady-state versions of the model (Section
2). In Section 3 we describe the results of a series of numerical simulations using the full model and the reduced models to
explore the interactions between accumulation, advection, grain size, and compaction. We discuss our results in Section 4 and
summarize conclusions and our outlook for future work in Section 5.

## 2 Methods


In this section we describe the model equations and boundary conditions, nondimensionalization of the model and the numerical methods used to solve the equations. We then describe a reduced, steady-state ordinary differential equation (ODE) model
that will help us examine the accumulation dependence of steady-state firn thickness.

### 2.1 Model equations and boundary conditions

We consider a one-dimensional, isothermal column of firn and ice. The model describes the coupled spatial and temporal
evolution of five properties of the firn, all defined in a bulk sense (i.e., considering a spatial scale much larger than the grain
size): porosity, vertical normal stress, vertical velocity, grain size, and age (Table 1). Unlike most previous firn compaction
modelling (Lundin et al., 2017) we use an Eulerian reference frame (Case and Kingslake, 2021). The vertical coordinate $z$
moves with the ice surface and increases downwards, $z = 0$ denotes the ice sheet surface, and $z = z_b$ denotes the lower limit of
the model domain. At any time $t$, the total thickness $h$ of the model domain is $h(t) = z_b$.





Porosity is defined as $\phi = 1 - \rho/\rho_i$, where $\rho$ is the depth-dependent density and $\rho_i$ is the density of ice, assumed constant (918 kg m$^{-3}$). While in reality firn temperatures vary seasonally near the surface, for simplicity we assume that the temperature is equal to the surface temperature $T_s$ everywhere. Table 2 summarizes the physical properties assumed constant in the model. Following Arthern et al. (2010), we describe firn compaction with a viscous constitutive relation:

$$\frac{D\phi}{Dt} = k_c \text{sign}(\sigma)|\sigma|^n \phi^m (1-\phi) f_T(T_s) f_r(r), \tag{2}$$

where $\sigma$ is the vertical normal stress (following the convention that compressive stresses are negative); $k_c$, $n$ and $m$ are constants; and $f_T$ and $f_r$ are functions of $T_s$ and grain radius $r$, respectively (Arthern et al., 2010). The sign function returns the sign of its argument. The material derivative is defined by $D/Dt = \partial/\partial t + w\partial/\partial z$, where $w$ is the vertical velocity of the ice and firn relative to ice-sheet surface, defined as positive downwards. Assuming a linear rheology, with linear dependence

on $\phi$, ($n = m = 1$), Arthern et al. (2010) found that $k_c = 9.2 \times 10^{-9}$ kg$^{-1}$ m$^3$ provided a reasonable fit to field observations of firn compaction and we adopt this value here. Following previous studies (Herron and Langway, 1980; Stevens et al., 2020), we adopt an Arrhenius relation for $f$,

$$f_T = \exp[-E_c/RT_s], \tag{3}$$

where $E_c$ is the activation energy for compaction (60 kJ mol$^{-1}$) and $R$ is the gas constant (8.3 J mol$^{-1}$ K$^{-1}$). Following

Arthern et al. (2010), we adopt $f_r(r) = 1/r^2$, consistent with Nabarro-Herring creep by diffusion through the crystal lattice. Combining this expression, Eq. 2, Eq. 3 and the definition of the material derivative yields an evolution equation for $\phi$:

$$\frac{\partial\phi}{\partial t} = k_c \frac{\text{sign}(\sigma)|\sigma|^n \phi^m (1-\phi)}{r^2} \exp[-E_c/RT_s] - w\frac{\partial\phi}{\partial z},$$
$$\phi(0) = \phi_s, \tag{4}$$

where $\phi_s$ is a prescribed surface porosity. The vertical gradient of overburden stress $\sigma$ is given by

$$\frac{\partial\sigma}{\partial z} = -\rho_i g(1-\phi),$$
$$\sigma(0) = 0, \tag{5}$$

where $g$ is acceleration due to gravity (9.8 m s$^{-2}$). Ignoring horizontal strain (Horlings et al., 2021; Case and Kingslake, 2021), mass conservation requires

$$\frac{D\phi}{Dt} = (1-\phi)\frac{\partial w}{\partial z} \tag{6}$$

Combining this with Eq. 4 provides an expression for the vertical gradient of $w$,

$$\frac{\partial w}{\partial z} = k_c \frac{\text{sign}(\sigma)|\sigma|^n \phi^m}{r^2} \exp[-E_c/RT_s],$$
$$w(0) = \frac{b}{1-\phi_s}, \tag{7}$$





**Table 2.** Physical constants.

| constant | description | value |
|---|---|---|
| $c$ | specific heat capacity of ice | 2.0 kJ kg$^{-1}$ K$^{-1}$ |
| $E_c$ | activation energy for compaction | 60 kJ mol$^{-1}$ |
| $E_g$ | activation energy for grain growth | 42 kJ mol$^{-1}$ |
| $g$ | acceleration due to gravity | 9.81 m s$^{-2}$ |
| $G$ | geothermal heat flux | 50 mW kg m$^{-1}$ |
| $k_c$ | compaction coefficient[a] | 9.2×10$^{-9}$ kg$^{-1}$ m$^3$ |
| $k_a$ | grain growth coefficient[b] | 1.3×10$^{-7}$ m s$^{-1}$ |
| $k_g$ | modified grain growth coefficient[c] | $k_a/r_f^2$ |
| $r_s^2$ | saturation grain size | 10$^{-4}$ m$^2$ |
| $R$ | universal gas constant | 8.3 J mol$^{-1}$ K$^{-1}$ |
| $\rho_i$ | ice density | 918 kg m$^{-3}$ |

[a] assumes $n = m = 1$,

[b] from Arthern et al. (2010),

[c] modified from $k_a$ to account for addition of $(r_f^2 - r^2)$

in Eq. 8.

where $b$ is the rate of accumulation of snow in units of ice-equivalent per unit time (i.e. the depth the snow that accumulates in

each unit time would have if it had the density of ice). The upper boundary condition on $w$ is motivated by the fact that $w$ is

the velocity relative to the ice surface. At the surface this is determined by the accumulation rate and the surface porosity.

We follow Arthern et al. (2010) in describing grain-size evolution as independent of stress and obeying an Arrhenius temperature dependence. This is referred to as normal or static grain growth (e.g., Gow, 1969; Alley and Woods, 1996; Jun et al.,

1998). However, we extend this model with the recognition that normal grain growth will not continue indefinitely, but will

eventually be significantly counteracted by flow-induced recrystallization and polygonization (e.g., Alley, 1992; Duval and

Castelnau, 1995; Mathiesen et al., 2004; Roessiger et al., 2011). We modify Arthern et al.'s grain growth equation to include

this effect in a simplistic way and adapt it to our Eulerian framework as follows:

$$\frac{\partial r^2}{\partial t} = k_g \exp[-E_g/RT_s](r_f^2 - r^2) - w \frac{\partial r^2}{\partial z},$$

$$r^2(0) = r_s^2. \tag{8}$$

where $E_g$ is the activation energy for grain growth (42 kJ mol$^{-1}$; Arthern et al., 2010), $r_f$ is a saturation grain radius, and

$r_s$ is the surface grain radius. Given that this expression describes the evolution of the square of the grain radius, hereafter we

refer to $r^2$ as the grain size and $r_f^2$ and the saturation grain size. We conservatively estimate $r_f^2 = 10^{-2}$ m$^2$. This is considered

a conservative estimate because it is low compared to observed saturation grain sizes (e.g., Mathiesen et al., 2004; Roessiger

et al., 2011) and because later we show that even with this lower-end estimate of $r_f^2$, grain-size saturation within the firn layer

is unlikely across a range of climates. The constant $k_g$ is defined by modifying Arthern et al.'s grain growth coefficient, $k_a =$





$1.3 \times 10^{-7}$ m$^2$ s$^{-1}$, to account for our addition of $(r_f^2 - r^2)$ in Eq. 8: $k_g = k_a/r_f^2$. For simplicity, Eq. 8 neglects the impact of impurities or microstructure on grain growth (e.g., Alley and Woods, 1996; Jun et al., 1998; Roessiger et al., 2011).

Although is has no impact on firn thickness, we include the following evolution for the age of the firn and ice $A$ to aid future work on the ice-gas age offset (e.g., Buizert et al., 2021):

$$\frac{\partial A}{\partial t} = 1 - w\frac{\partial A}{\partial z}$$

$$A(0) = 0. \tag{9}$$

Finally, we use domain-wide mass conservation (Appendix A) to derive a kinematic condition for the time evolution of the thickness of the domain:

$$\frac{\partial h}{\partial t} = w(z_b) - \frac{b}{1 - \phi_b}, \tag{10}$$

which indicates that the lower limit of the domain moves due to any imbalance between the ice-equivalent accumulation rate $b$
(with units of length per time) and the velocity at the lower surface.

Equations 4–10 complete the model. It describes how 5 variables – $\phi$, $\sigma$, $w$, $r^2$, $A$, and $h$ – vary in response to prescribed surface porosity $\phi_s$, surface grain size $r_s^2$, surface temperature $T_s$ and accumulation rate $b$.

## 2.2 Nondimensionalization

We define scales as follows:

$\phi = \phi^*\phi_0, \quad \sigma = \sigma^*\sigma_0, \quad w = w^*w_0,$
$r^2 = r^{2*}r_0^2, \quad A = A^*A_0, \quad h = h^*h_0, \quad b = \beta b_0, \quad z = z^*z_0, \quad t = t^*t_0, \tag{11}$

where symbols with asterisks represent scaled variables, coordinates or parameters, and the zero subscripts denote scales. We use $\beta$ to denote the nondimensional accumulation rate to distinguish this input parameter from the model variables. We scale $w$ by the accumulation rate scale, $w_0 = b_0$, which we prescribe later, and define $t_0$ as the characteristic transit time of material
through the domain, $t_0 = z_0/w_0$. We set $z_0 = h_0 = 100$ m and $\phi_0 = 1$.

Substituting scales into Eq. 4 (dropping asterisks for clarity) yields

$$\frac{\partial \phi}{\partial t} = -\frac{1}{\alpha}\frac{|\sigma|^n\phi^m(1-\phi)}{r^2} - w\frac{\partial \phi}{\partial z},$$
$$\phi(0) = \phi_s, \tag{12}$$

where we have used the fact that $\sigma \leq 0$ (Eq. 5) and introduced the nondimensional parameter $\alpha$, which controls the relative
contributions of compaction and advection:

$$\alpha = \frac{r_0^2}{k_c t_0 \sigma_0^n \exp\left[-E_c/RT_s\right]}. \tag{13}$$





Defining $\sigma_0 = \rho_i g z_0$, Eq. 5 becomes

$$\frac{\partial \sigma}{\partial z} = -(1 - \phi),$$
$$\sigma(0) = 0 \tag{14}$$

and Eq. 7 becomes

$$\frac{\partial w}{\partial z} = -\frac{1}{\alpha} \frac{|\sigma|^n \phi^m}{r^2},$$
$$w(0) = \frac{\beta}{1 - \phi(z_s)}. \tag{15}$$

Equating terms in Eq. 8 yields

$$r_0^2 = \frac{k_g z_0 r_f^2}{b_0} \exp\left[-E_g / R T_s\right] \tag{16}$$

and the nondimensional grain size evolution equation,

$$\frac{\partial r^2}{\partial t} = (1 - \delta r^2) - w \frac{\partial r^2}{\partial z},$$
$$r^2(0) = r_s^2, \tag{17}$$

where $\delta = r_0^2 / r_f^2$. Defining $A_0 = t_0$ leaves the age Equation (9) cosmetically unchanged,

$$\frac{\partial A}{\partial t} = 1 - w \frac{\partial A}{\partial z},$$
$$A(0) = 0. \tag{18}$$

Finally, Eq. 10 becomes

$$\frac{\partial h}{\partial t} = w(z_b) - \frac{\beta}{1 - \phi_b}. \tag{19}$$

## 2.3  Parameter values

Table 3 shows the values of model scales and dimensionless parameters corresponding to three climates with high ($b_0 = 1$ m
yr$^{-1}$, $T_s = 0$ °C), intermediate ($b_0 = 0.1$ m yr$^{-1}$, $T_s = -20$°C), and low ($b_0 = 0.01$ m yr$^{-1}$, $T_s = -40$°C) accumulation and
surface temperatures. The timescale, $t_0$, is controlled only by $b_0$ and the prescribed depth scale, $z_0$. It varies from 100 years in
the high accumulation climate to 10,000 years in the low accumulation climate.

The dimensionless number $\alpha$ describes the relative contributions of firn advection and compaction to the evolution of $\phi$ for
our choice of $z_0$ (100 m); higher $\alpha$ indicates slower compaction. Its dependence on $T_s$ and $b_0$ is controlled by the competing
dependencies of grain growth, advection and compaction, on $T_s$ and $b_0$. Combining Eqn 13 and Eqn 16 yields

$$\alpha = \frac{k_g}{k_c \sigma_0^n} \exp[(E_c - E_g) / R T_s]. \tag{20}$$





**Table 3.** Surface temperatures, scales, and nondimensional parameters corresponding to three climatic settings: (1) high accumulation and surface temperature (e.g., a mountain glacier in a maritime climate), (2) intermediate temperature and accumulation (e.g., near-coastal Antarctica), and (3) low temperature and accumulation (e.g., interior East Antarctica).

| parameter/scale | description [units] | high | intermediate | low |
|---|---|---|---|---|
| $b_0$ | accumulation scale [m a$^{-1}$] | 1 | 0.1 | 0.01 |
| $T_s$ | surface temperature [K] | 273 | 253 | 233 |
| $r_0$ | grain radius scale [m$^2$] | $3.8 \times 10^{-6}$ | $8.8 \times 10^{-6}$ | $1.6 \times 10^{-5}$ |
| $z_0$ | vertical scale [m] | 100 | 100 | 100 |
| $w_0$ | velocity scale [m a$^{-1}$] | 1 | 0.1 | 0.01 |
| $t_0$ | time scale [a] | 100 | 1000 | 10000 |
| $\sigma_0$ | overburden stress scale [Pa] | $9.0 \times 10^{-5}$ | $9.0 \times 10^{-5}$ | $9.0 \times 10^{-5}$ |
| $\alpha$ | compaction number [-] | 0.044 | 0.082 | 0.170 |
| $\delta$ | grain size saturation ratio [-] | 0.038 | 0.088 | 0.16 |

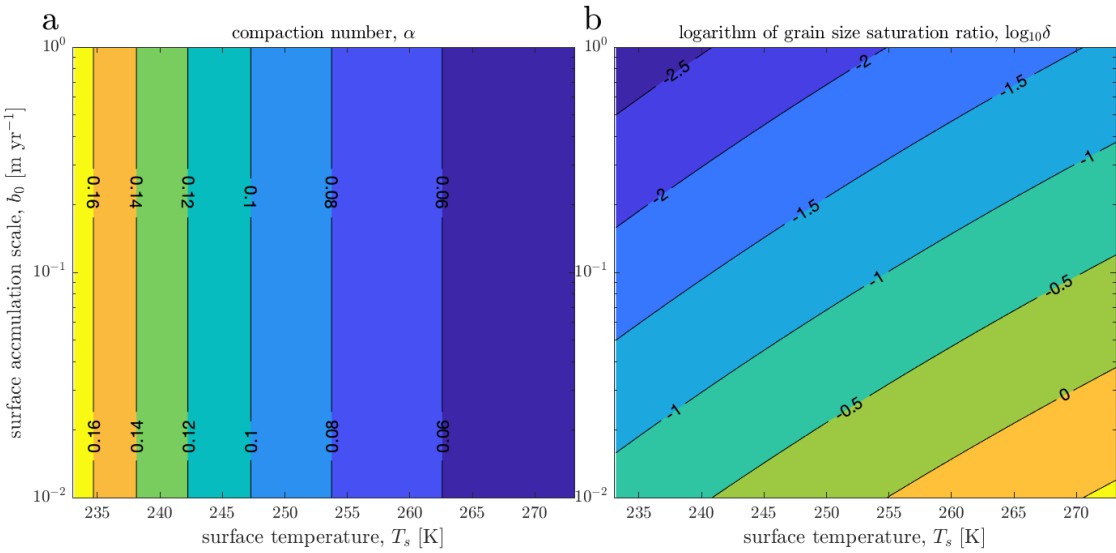

**Figure 1.** Two model parameters, (a) the compaction number $\alpha$ and (b) the base-10 logarithm of the grain size saturation ratio $\delta$, evaluated for a range of accumulation scales $b_0$ and surface temperatures $T_s$.





This shows that while higher temperatures tend to accelerate compaction directly (the first term in the exponent), this is counteracted by the effect of increasing temperatures on grain size (the second term in the exponent); higher $T_s$ leads to faster grain growth (Eq. 16), which tends to slow down compaction (Eq. 12). However, because $E_c > E_g$ the net effect of increasing $T_s$ is faster compaction (decreased $\alpha$). In contrast, $\alpha$ is independent of $b_0$ because the impact of $b_0$ on $D\phi/Dt$ (reflected by the $t_0$ in the denominator of Eq. 13) is balanced by the impact of $b_0$ on grain size (reflected by the $r_0^2$ in the numerator of Eq. 13 and the $b_0$ in the denominator of Eq. 16). While competition between the effects of grain size evolution and advection manifests here purely in terms of the scales and nondimensional parameters of the scaled model, we will discuss the same competition in more detail later when it reappears while considering the effect of varying the nondimensional inputs to the model between simulations (specifically, $r_s^2$ and $\beta$). Given this relationship between $\alpha$ and $T_s$, $\alpha$ increases by a factor of four between the high and low temperature climates (Table 3; Figure 1). However, even in the low temperature climate, $\alpha < 1$, indicating that compaction is large compared to advection and that firn will usually closely approach zero at depth in our simulations.

The grain-size saturation parameter $\delta$ provides a measure of how important the addition of $(r_f^2 - r^2)$ in Eq. 8 is for the evolution of $r^2$, i.e. how closely the grain size will approach it saturation value $r_f^2$ within the firn layer. The dependence of $\delta$ on $T_s$ and $b_0$ is controlled by the grain size scale, $r_0$ (Eq. 16), which is a first-order estimate of the growth in grain size within the firn layer in the absence of grain growth saturation. $r_0$ increases with $T_s$ because grain growth increases with temperature, and decreases with $b_0$ because higher accumulation decreases the time available for grain growth before firn advects through the firn layer. For the three climates considered in Table 3, the accumulation dependence is the larger contributor, and $\delta$ is largest in the low-temperature, low-accumulation climate. Figure 1b shows that $\delta$ only reaches unity in relatively high-temperature $(T_s > 255K)$, low-accumulation $(b_0 > 0.04\text{m yr}^{-1})$ conditions. Considering the observed correlation between accumulation and temperature over ice sheets (e.g., Dalaiden et al., 2020), this combination of conditions are likely to be rare, indicating that grain size is unlikely to saturate within the firn layer and that $\delta$ can safely be neglected when necessary.

## 2.4  Numerics

Equations 12, 14, 15, 17, 18 and 19 describe our full nondimensional firn compaction model. We solve the equations numerically using the method of lines. We use a change of coordinates (Appendix B) to account for the temporally varying domain length. The method of lines involves discretizing the model domain in space into $N - 1$ grid steps connected at nodes and forming a coupled set of ordinary differential equations (ODEs) which describe the time evolution of the model variables. See Appendix B for more details. Unless otherwise stated, we use a grid spacing of $\Delta z = 0.01$. The ODEs are solved simultaneously using the MATLAB ODE solver, ode15s. This solver finds optimal time steps dynamically with user configurable absolute and relative error tolerances. We set these tolerances to $10^{-8}$.

All simulations use the following initial conditions:

$$\phi = (1-z)\phi_s; \quad r^2 = z + r_s^2; \quad h = 1; \quad A = z. \tag{21}$$

Unless otherwise stated, simulations continue until a steady state is reached, as detected when $|\partial\phi/\partial t| < 10^{-5}$ everywhere. See the code availability section for access to model code and figure plotting scripts.





## 2.5 A reduced, steady-state model

As well as presenting numerical solutions of the full model, we utilize a simplified, steady-state model consisting of a set of coupled ODEs. The purpose of the ODE model is to allow us to test our numerical solutions of the full model and to act as a starting point for several further simplifications designed to clarify the interdependence of firn thickness, porosity and grain size.

Equating the time derivatives in Equations 12, 17, and 18 to zero, rearranging, and gathering the results with Equations 14 and 15 yields

$$
\begin{aligned}
\frac{d\phi}{dz} &= -\frac{|\sigma|^n \phi^m (1-\phi)}{\alpha w r^2} \\
\frac{d\sigma}{dz} &= -(1-\phi) \\
\frac{dw}{dz} &= -\frac{|\sigma|^n \phi^m}{\alpha r^2} \\
\frac{dr^2}{dz} &= \frac{1-\delta r^2}{w} \\
\frac{dA}{dz} &= \frac{1}{w},
\end{aligned}
\tag{22}
$$

with the following boundary conditions :

$$
\phi(0) = \phi_s; \quad \sigma(0) = 0; \quad w(0) = \frac{\beta}{1-\phi_s}; \quad r^2(0) = r_s^2; \quad A(0) = 0.
\tag{23}
$$

Below we present the results of solving this model, and simplifications of it, using MATLAB's ODE solver, ode45, with the same grid spacing as used for the full model (previous section) and absolute and relative error tolerances of $10^{-10}$.

## 3 Results

### 3.1 Comparing the full and ODE model results

Figure 2 displays steady state profiles of $\phi$, $\sigma$, $w$, $r^2$, and $A$, resulting from solving our time-dependent firn compaction model following the methods described in Section 2.4. The non-dimensional accumulation rate is $\beta = 1$ and values for model scales and parameters are noted in the figure caption.

Porosity closely approaches a steady state ($|\partial\phi/\partial t| < 10^{-5}$ everywhere) after a nondimensional time of 0.83 (830 years). The steady-state vertical velocity $w$ is approximately equal to $\beta$ over $0.5 \leq z \leq 1$ and increases in magnitude towards the surface, where it reaches $\beta/(1-\phi)$. The porosity $\phi$ is zero in the lower region and increases towards its prescribed value at the surface. Pointing to limitations in the model that we discuss later, there is an inflection point in $\phi$ at $z = 0.212$ (21.2 m below the surface) and, because the overburden pressure is zero at the surface, the gradient of $\phi$ is also equal to zero at the surface ($\partial\phi/\partial z = 0$; Figure 2a; Eq. 12). The age $A$, $\sigma$, and the grain size $r^2$ increase approximately linearly with depth, deviating from linear where the gradients in the $w$ and $\phi$ deviate from zero.

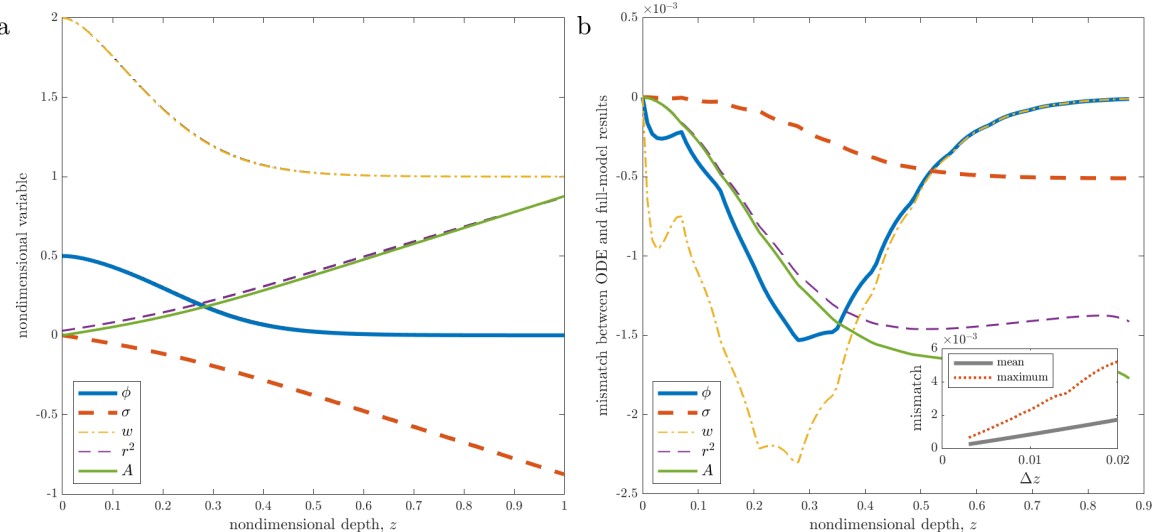

**Figure 2.** (a) A solution to the reduced ODE model overlaid on a steady-state solution to the full compaction model. Both simulations use $\Delta z = 0.01$, $r_s^2 = 0.029$, $b_0 = 0.1$ m a$^{-1}$, $\beta = 1$, $\phi_s = 0.5$, $T_s = 253.15$ K, $r_f^2 = 10^{-2}$ m$^2$, and $n = m = 1$, which yields $\alpha = 0.082$ and $\delta = 0.088$. The two sets of curves are indistinguishable at the scale of the plot. (b) Mismatch between each variable computed with each model, as functions of $z$. Inset shows the mean and maximum mismatch (over all $z$ and all five variables) as functions of $\Delta z$.

Plotted over the full-model results in Figure 2a are results from the ODE model (Section 2.5) computed using the same scales and parameters. The two sets of results are indistinguishable at the scale of the plot. Figure 2b shows the mismatch between

the two sets of results for each variable. The mean absolute difference between the two solutions across the five variables is $8.3 \times 10^{-4}$ and the maximum difference is $2.3 \times 10^{-3}$. Mean and maximum differences as percentages of the full-model values are 0.55% and 3.67%, respectively. The inset in Figure 2b shows how the mean and maximum mismatch vary between simulations that used a range of grid spacings, $\Delta z$. The fact that the mismatch between the results is small ($\ll 1$) and that the mean and maximum mismatches approach zero as the grid spacing decreases gives us confidence that our numerical method

recovers the full model's steady state with sufficient accuracy for the purposes of the following analysis.

## 3.2 Uniform and constant grain size

To better understand the accumulation dependence of the thickness of the firn layer, we consider a simple case with uniform and constant grain size, $r^2(z,t) = r_s^2$. Figure 3a plots steady state porosity profiles simulated using the full model. Each simulation used the same boundary conditions and parameters values (following the 'intermediate' scenario in Table 3), but a different

nondimensional accumulation rate, $\beta$. In all simulations the grain size is initiated as uniform and equal to the surface grain size, and is not updated during the simulation.





**Figure 3.** Accumulation dependence in the absence of grain-size evolution in three different models. The left panels show porosity $\phi$ as a function of depth $z$. The right panels show velocity $w$ as a function $z$. Each row shows the results of 20 simulations each using a different accumulation rate $\beta$, which varies linearly between 0.5 and 10. The arrows show the direction of increasing $\beta$. All simulations used $b_0 = 0.1$ m a$^{-1}$, $\phi_s = 0.5$, $T_s = 253.15$ K, $r_f^2 = 0.01$ m$^2$, $n = m = 1$. They also all prescribe $r^2$ equal to its surface value $r_s^2 = 0.029$ everywhere (corresponding to a dimensional grain radius of $\sqrt{0.029 r_0^2} = 0.5$ mm). Scales and other parameter values are in the *intermediate* column in Table 3. (a) and (b): Steady-state solutions of the full model (Section 2.4). (c) and (d): Solutions to Eq. 24, which assumes $r^2(z) = r_s^2$, $\sigma(z) = -z$. (e) and (f): Solutions to Eq. 25, which assumes $r^2(z) = r_s^2$, $\sigma(z) = -z$, and $w(z) = -\beta$. The insets in (a), (c) and (e) show the dependence of firn thickness $z_{830}$ on $\beta$ for all three sets of simulations (dotted curves) with each solid curve corresponding to the results shown in each row.





In all simulations, higher accumulation leads to thicker firn; $z_{830}$, the nondimensional depth corresponding to a porosity of $\phi = 1 - 830/\rho_i = 0.096$, increases sub-linearly with the accumulation rate $\beta$ (inset, Figure 3a). Firn thickness is controlled by a competition between porosity advection and compaction. The positive $z_{830}$-$\beta$ relationship is due to the increased accumulation

leading to increased downward advection of higher porosity firn (second term on the right of Eq. 12; Figure 3b), which the corresponding increase in compaction rate (the first term on the right of Eq. 12) is insufficient to balance. Therefore, the result of increasing $\beta$ is that a given parcel of firn does not reach the bubble-close-off density of 830 kg m$^{-3}$ until it has reached a greater depth.

Simplifying the ODE model helps to demonstrate this behavior and will assist with contrasting it to the case when the

grain size is allowed to evolve, presented in the next section. We start by ignoring the age equation, which has no effect on the $\beta$ dependence of firn thickness, and assuming $r^2(z) = r_s^2$. The results presented in Figure 2a motivate two additional simplifications. Firstly, recognizing that $\sigma$ is approximately linear, we substitute $\sigma = -z$ into Equations 22.1 and 22.2, reducing the ODE model to

$$\frac{d\phi}{dz} = -\frac{z^n \phi^m (1 - \phi)}{\alpha w r_s^2},$$

$$\frac{dw}{dz} = -\frac{z^n \phi^m}{\alpha r_s^2}, \tag{24}$$

with $\phi(0) = \phi_s$ and $w(0) = \beta/(1 - \phi_s)$. Figures 3c and 3d plot solutions to Eq. 24. The results retain the sub-linear positive relationship between $\beta$ and $z_{830}$ (inset, Figure 3c). The second simplification ignores the impact of compaction on $w$ by assuming $w(z) = \beta$, which reduces Eq. 24 to

$$\frac{d\phi}{dz} = -\frac{z^n \phi^m (1 - \phi)}{\alpha \beta r_s^2}, \tag{25}$$

with $\phi(0) = \phi_s$. Figures 3e and 3f plot solutions to this equation. These too retain the sub-linear positive relationship between $\beta$ and $z_{830}$. Because $\beta$ is in the denominator in Eq. 25, higher accumulation leads to thicker firn by decreasing the vertical gradient of $\phi$. It achieves this by increasing downward advection. We know this because $\beta$ appeared in this equation via our simple assumption of $w(z) = \beta$.

Ignoring the impact of porosity on $\sigma$ to reach Eq. 24 and on both $\sigma$ and $w$ to reach Eq. 25 renders these reduced models highly

simplistic representations of firn compaction. Nonetheless, the fact that each progressively simpler model shares a qualitatively similar relationship between $\beta$, $z_{830}$ and $w$ indicates that even the simplest ODE model captures the essence of the physics underlying these relationships; specifically, increased accumulation leads to increased downward advection of high porosity firn.

### 3.3 Grain size evolution

Next we consider how grain-size evolution affects the dependence of firn thickness on accumulation. Figure 4 displays steady-state results of three sets of simulations using the full model. In contrast to the simulations discussed in the previous section, in these simulations the grain size is allowed to evolve in space and time through grain growth (first term on the right of Eq. 17)





and advection (second term on the right of Eq. 17). The surface grain size $r_s^2$ varies between the three sets of simulations and the accumulation rate $\beta$ varies between members of each set of simulations. All other model parameters are uniform across

simulations and equal to those used to produce the results displayed in Figure 2 and 3a.

In all simulations, just as in the previous section where $r^2$ did not evolve, firn thickness $z_{830}$ increases sub-linearly with accumulation rate $\beta$. However, the strength of this dependence decreases as the surface grain size $r_s^2$ decreases (Figures 4). This can be observed in the variability in the spread of the porosity profiles in Figure 4a, 4b and 4c. Quantitatively, when surface grain size is relatively large ($r_s^2 = 0.1$), the gradient of $z_{830}$ with respect to $\beta$ has a mean value of 0.075 (Figure 5) and varies

from 0.26 at $\beta = 0.1$, to 0.048 at $\beta = 10$ (inset, Figure 4c). In contrast, when surface grain size is relatively small ($r_s^2 = 0.001$), the gradient of $z_{830}$ with respect to $\beta$ has a mean value of 0.0050 (Figure 5) and varies from 0.023 at $\beta = 0.1$ to 0.0040 at $\beta$ = 10 (inset, Figure 4a). Figure 6a shows how $z_{830}$ depends on both parameters together. Over this range of parameter values, this mutual dependence is approximately symmetric; increasing $\beta$ increases the dependence of $z_{830}$ on $r_s^2$ and increasing $r_s^2$ increases the dependence of $z_{830}$ on $\beta$.

We turn to the ODE model to understand the dependency of the accumulation sensitivity on surface grain size. Starting with Eq. 22, instead of $r(z) = r_s$ (which leads to Eq. 24), we assume $\delta = 0$ (as motivated by the discussion in Section 2.3). Additionally assuming $\sigma = -z$ as before yields

$$\frac{d\phi}{dz} = -\frac{z^n \phi^m (1-\phi)}{\alpha w r^2},$$
$$\frac{dw}{dz} = -\frac{z^n \phi^m}{\alpha r^2},$$
$$\frac{dr^2}{dz} = \frac{1}{w}, \tag{26}$$

with $\phi(0) = \phi_s$, $w(0) = \beta/(1-\phi_s)$, and $r^2(0) = r_s^2$. Figure 6b shows the dependence of $z_{830}$ on $\beta$ and $r_s^2$ computed using Eq. 26. Qualitatively, the relationship between these three quantities is the same as found with the full model (Figure 6a).

To simplify the model further we assume $w = \beta$, then integrate Eq. 26.3, rearrange the result, and substitute it into Eq. 26.1, yielding

$$\frac{d\phi}{dz} = -\frac{z^n \phi^m (1-\phi)}{\alpha} \overbrace{\frac{1}{\beta}}^{\text{from } w} \overbrace{\frac{1}{\left(\frac{z}{\beta} + r_s^2\right)}}^{\text{from } r^2} = -\frac{z^{n-1}\phi^m(1-\phi)}{\alpha\left(1 + \frac{\beta r_s^2}{z}\right)}. \tag{27}$$

Figure 6c shows the dependence of $z_{830}$ on $\beta$ and $r_s^2$ computed using this simple model. The qualitative agreement between the three panels in Figure 6, each resulting from a progressively simpler description of firn compaction with grain-size evolution, indicates that insight into the full model's dependence on $\beta$ and $r_s^2$ can be gained from the simplest model.

Note that we recover Eq. 25 if we assume $\beta r_s^2 \gg 1$ in Eq. 27, which is equivalent to neglecting grain-size evolution. There-
fore, comparing Equations 25 and 27 shows that $\partial\phi/\partial z$ is always smaller when the grain size is allowed to evolve, which leads to a thicker firn layer. The explanation is that allowing grain size to evolve leads to larger grains, which slows compaction (e.g., Eq. 4).

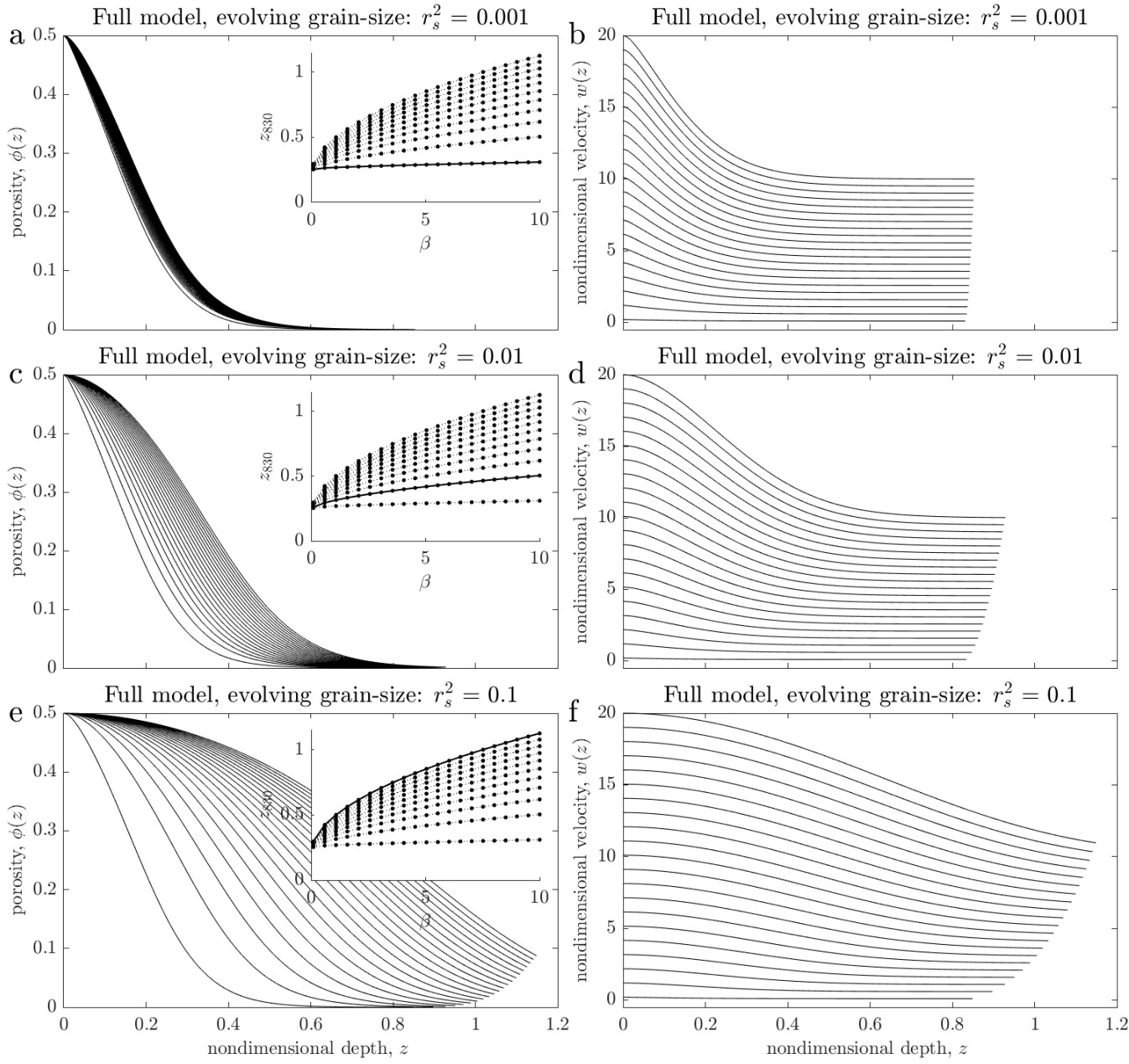

**Figure 4.** Accumulation dependence of the full model, with a grain size that evolves. The layout is similar to Figure 3, except that only results from the full model are shown and the rows display results from three sets of selected simulations from among 11 sets of simulations, each using a different value of the surface grain size, $r_s^2$. Across each set of simulations the accumulation varies linearly between 0.1 and 10. The insets in (a), (c) and (e) show the dependence of firn thickness $z_{830}$ on accumulation rate $\beta$ for all 11 sets of simulations (dotted curves) with each solid curve corresponding to the results shown in each row.



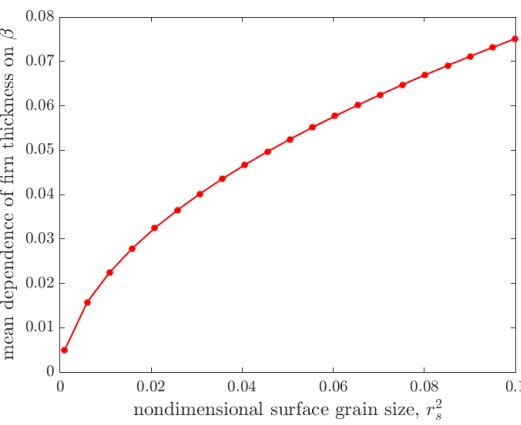

**Figure 5.** The dependence of firn thickness $z_{830}$ on accumulation rate $\beta$ and how this varies with surface grain size, $r_s^2$. These values were computed as the gradients of the curves in the insets of Figure 4 (and from the results of 10 additional simulations), which each used a different values of $r_s^2$. The gradients were computed using linear least-squares regression.

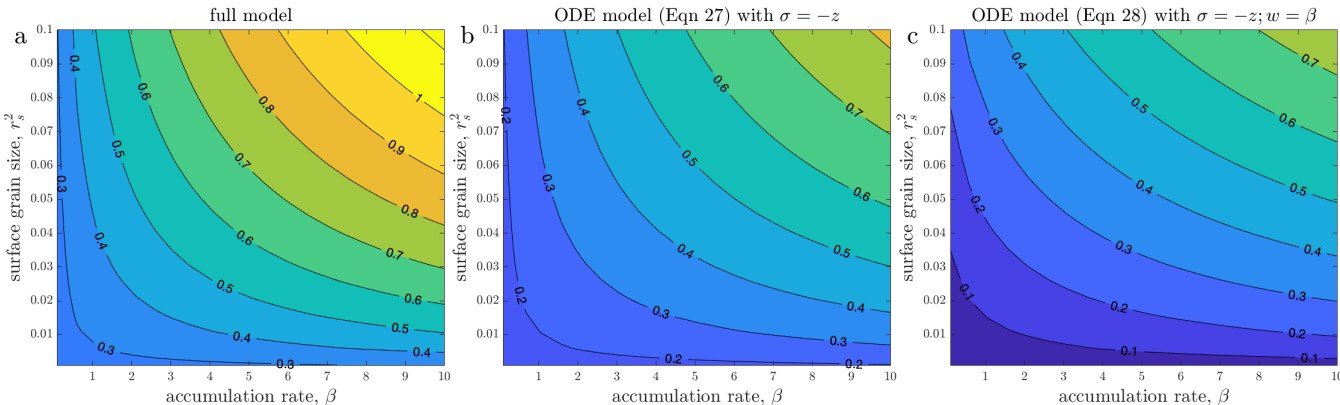

**Figure 6.** Firn thickness $z_{830}$ as a function of surface grain size $r_s^2$ and accumulation rate $\beta$ (all nondimensional) from (a) the full model using the same parameters as used to produce Figure 4, (b) a simplified version of the ODE model that assumes $\delta = 0$ and $\sigma = -z$ (Eq. 26), and (c) a further simplification of the ODE model that assumes $\delta = 0$, $\sigma = -z$, and $w = -\beta$ (Eq. 27).





The overbraces in Eq. 27 indicate the origin of two competing $\beta$-dependencies. The velocity $w$ introduces an inverse dependence on $\beta$ due, again, to faster downward advection of high-porosity firn. However, this is partially compensated for by

the dependence of $\partial\phi/\partial z$ on grain size $r^2$, which introduces another dependence on $\beta$. Recall that grain size increases with depth (green dashes curve in Figure 2). Therefore, because grain size is advected with the firn as it moves downwards, and because $\beta$ controls the rate of advection, a larger $\beta$ leads to a smaller $r^2$ everywhere. Because smaller-grained firn compacts more easily (Eq. 12), this increase in advection leads to faster compaction and reduces firn thickness. The net result of the inverse dependence of $\partial\phi/\partial z$ on $\beta$ from the advection of $\phi$ and its positive dependence on $\beta$ from the advection of $r^2$ is that

firn thickness increases with $\beta$, however, this dependence is not as strong as it is in the case when $r^2$ does not evolve with depth (Section 3.2).

The denominator on the right of Eq. 27 indicates that the overall dependence of $\phi$ on $\beta$ increases with $r_s^2$. This is consistent with the results of simulations using the full model (Figures 4 and 5). In fact, if surface grain size is sufficiently small that we can assume $r_s^2 = 0$ (or more precisely if $\beta r_s^2 \ll 1$), Eq. 27 has no dependence on $\beta$. The explanation is that when $r_s^2 = 0$,

the grain size profile is simply $r^2(z) = z/\beta$, i.e. linearly increasing with depth at a rate inversely proportional to $\beta$. Combined with the linear dependence of compaction rate on $r^2$, this means that simulations with higher $\beta$ have lower grain size and therefore faster compaction. This effect exactly balances the effect of faster downward advection of porosity in simulations with higher $\beta$. In other words, the effect of increased porosity advection exactly balances the effect of increased grain-size advection when $r_s^2 = 0$. More generally, when $r_s^2 \neq 0$, these two effects do not balance exactly, but because $r^2 = r_s^2 + z/\beta$,

decreasing $r_s$ increases the relative importance of $\beta$ in determining $r_s^2$, which increases the size of the grain-size-advection effect and reduces the dependence of firn thickness on accumulation.

While the explanation for the relationship between $z_{830}$, $\beta$, and $r_s^2$ has been given in reference to a simplified ODE model (Eq. 27), the same competition between advection of grain size and advection of porosity operates in the full model. This is reflected in the numerical results shown in Figure 4, 5, and 6a. It can also be shown analytically that $z_{830}$ is independent

of $\beta$ when $r_s^2 = 0$ (Appendix C). Despite the vertical variation of velocity being more complex in the full model than in the ODE model (it is a function of grain size and porosity, rather than simply assumed constant), increasing $\beta$ still leads to faster advection of $\phi$, the effect of which is exactly balanced by increased downward advection of grain size when $r_s^2 = 0$.

### 3.4 Nonlinear stress dependence

All results presented above assumed a linear viscous rheology where compactive strain rates depend linearly on the overburden

stress ($n = 1$). To examine the effect of a nonlinear stress dependence ($n \neq 1$) on the relationship between $z_{830}$, $\beta$, and $r_s^2$, we performed additional simulations using the full model. The rheological parameters introduced in Eq. 4 were derived assuming $n = 1$ (Arthern et al., 2010). To allow for a more reasonable comparison between multiple simulations using $n \neq 1$, we introduce a modified compaction number, $\alpha' = \alpha(\sigma_0/4)^{1-n}$. This was derived by equating the strain rates (Eq. 15) resulting from an arbitrarily chosen intermediate stress of $\sigma_0/4$ computed using $n = 1$ and using $n \neq 1$. Our results depend only quantitatively

on this arbitrary choice. Figure 7 plots the dependence of firn thickness on accumulation rate and surface grain size using $n = 2$, 3 and 4, using corresponding values of $\alpha'$. Qualitatively the results are the same as the full-model results computed using





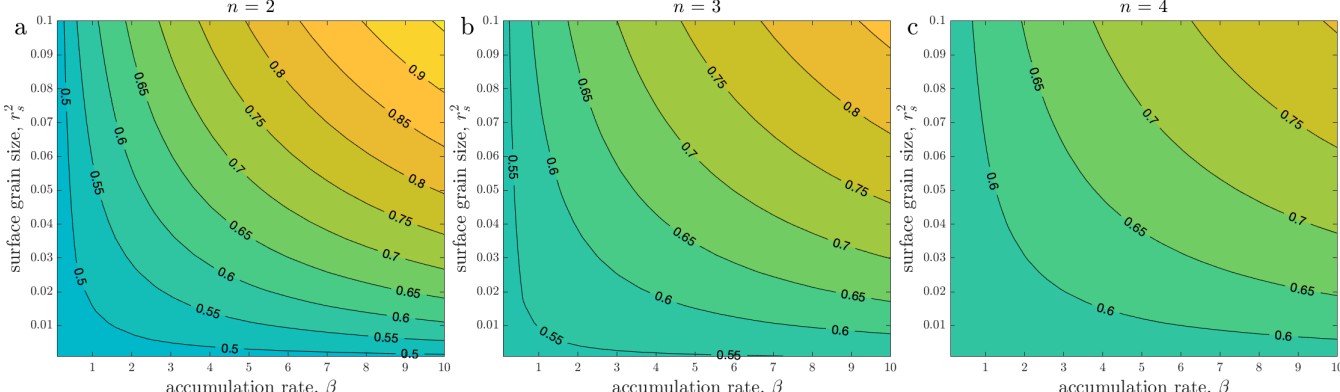

**Figure 7.** Firn thickness $z_{830}$ as a function of surface grain size $r_s^2$ and accumulation rate $\beta$ using the full model with (a) $n = 2$, (b) $n = 3$, and (c) $n = 4$ and corresponding values of $\alpha'$. All other parameters are the same as used for the simulations plotted in Figure 6. The color scales are the same as used in Figure 6.

$n = 1$ (Figure 6a); increasing surface grain size increases the dependence of firn thickness on accumulation rate. This indicates that the mechanisms relating $z_{830}, \beta$ and $r_s^2$ discussed above are independent of the stress dependence of compaction. This is consistent with Eq. 27, where the stress exponent does not effect the relationship between these quantities in the simple ODE
model. It is also consistent with the analysis in Appendix C of the full model.

### 3.5 Ice surface height change

All results presented above have been from steady states where the height of the ice sheet surface, represented in this model by the domain thickness $h$, has ceased changing significantly ($\dot{h} \approx 0$). This state is reached because the velocity at the bottom of the domain, which is the result of the prescribed upper-surface boundary condition on $w$ and the integrated compactive
strain rate (Eq. 15), has closely approached the prescribed accumulation (Eq. 19). Figure 8 displays results from a series of experiments in which the ice surface height instead continually increases (left panel) or decreases (right panel). This is implemented by increasing or decreasing the accumulation rate by 10% (i.e. multiplying the second term on the right of Eq. 19 by 1.1 or 0.9, respectively), while maintaining the upper-surface boundary condition on velocity: $w(z_b) = \beta/(1 - \phi(z_s))$. This simulates the scenario where the flow of the ice sheet is in equilibrium with an accumulation rate of $\beta$, but the climatically
controlled accumulation is larger or smaller than this value. Such a scenario is possible if the response time of the flow of the ice sheet is much larger than the time scale of climate variability. The result is that after a initial transient period, $h$ increases or decreases at a constant rate and the vertical variations of all model variables with respect to the surface remain constant, despite continuous surface height change.

Figure 8 shows the steady state $z_{830}$ resulting from these two sets of experiments as functions of accumulation rate $\beta$ and
the surface grain size $r_s^2$. Comparison between the two panels and between this figure and Figure 6a shows that steady state $z_{830}$ is larger when the ice surface height increases and smaller when the surface height decreases over time. The explanation





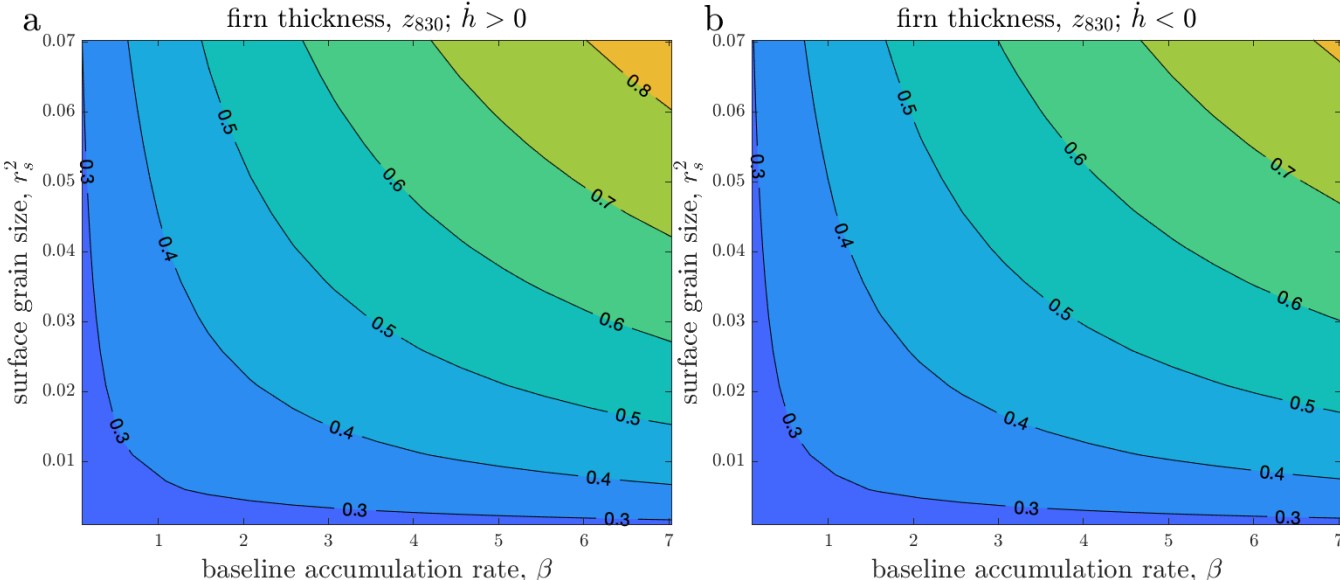

**Figure 8.** Steady-state firn thickness $z_{830}$ simulated with the full model during steadily (a) increasing and (b) decreasing ice-surface height and domain length $h$, as functions of surface grain size $r_s^2$ and the baseline accumulation rate $\beta$. Simulations use the same parameters as used to produce Figure 4 except that the baseline accumulation rate in Eq. 19 is multiplied by (a) 1.1 and (b) 0.9, to force continuous surface height change. Color scales are the same as used in Figures 6 and 7. Simulations terminate after a nondimensional time of $1/\beta$, which provides enough simulation time for the variation of $\phi$ with respect to the surface to reach a steady state.

is that the raising or lowering of the ice surface effectively increases or decreases, respectively, the advection of higher porosity density firn downwards. For example, in the case of continuous, steady surface-height increase, as a parcel of firn is buried and gradually compacts the surface moves upwards and by the time the parcel of firn reaches the bubble close-off porosity it has
reached a larger depth than it would have reached if the surface was stationary. Nonetheless, Figure 8 shows that qualitatively the relationships between $z_{830}, \beta$ and $r_s^2$ are unchanged from the $\dot{h} \approx 0$ cases considered in previous sections, indicating that the mechanisms relating these quantities discussed above also operate when the ice surface height is changing in time.

## 4 Discussion

We have described a firn compaction model that includes grain size evolution. What distinguishes it from most previous
models is that is uses an Eulerian reference frame, following Case and Kingslake's (2021) adaptation of Arthern et al.'s (2010) equations. Going further than Case and Kingslake (2021), we scaled the model equations and included grain-size saturation, which a scaling analysis suggested is generally negligible in firn. We also derived a simple ODE model from the full model, which can be used to simulate porosity, age, and grain size, when surface forcings change slowly enough that a steady state can be assumed.





We used these models to examine how accumulation affects firn thickness through its impact on the competing processes of compaction and advection. An Eulerian reference frame lends itself to this analysis as it allows us to compare terms describing both processes. We first considered the case when grain size is uniform and constant – which is the case considered by most previous firn models (Stevens et al., 2020) – then we allowed grain size to evolve through grain growth and grain-size advection (Arthern et al., 2010).

When grain size is kept uniform and constant, increasing accumulation increases downward advection. This is not balanced by the resulting increase in compactive strain rates and the net effect is that firn thickness increases sub-linearly with accumulation rate (Figure 3). An evolving grain size reduces both the steady-state firn thickness and the dependence of steady-state firn thickness on accumulation rate. Higher accumulation rate increases downward advection of lower porosity firn, increasing firn depth, but it also increases the downward advection of small-grained firn, which in this model compacts faster than larger-
grained firn. These two effects counteract each other, reducing the overall dependence of firn thickness on accumulation rate. We demonstrated this effect using numerical solutions of the full model and explained it using highly simplified versions of the steady-state ODE model.

     We showed that the extent to which grain size advection counteracts porosity advection increases as the surface grain size is decreased between simulations. Therefore, the sensitivity of firn thickness to accumulation rate increases with the surface
grain size, in this simple model. This is independent of the stress exponent in the firn constitutive relation, and of whether the ice surface height is increasing or decreasing at a steady rate. This is significant because if this relationship manifests in nature, then spatial and temporal variability in surface grain size driven by meteorological conditions will translate into spatial and temporal variability in the sensitivity of firn thickness to accumulation rates. Consideration of this effect could yield improvements to reconstructions of past climate that exploit modelled relationships between accumulation, bubble-close off
and stable isotope ratios (Buizert et al., 2021).

     We also considered the case when the grain size can be assumed to be zero at the surface (i.e., when $\beta r_s^2 \ll 1$). Under this assumption, the effects of porosity advection and of grain-size advection balance each other exactly and modelled firn thickness has no dependence on accumulation rate. Although this assumption may be unrealistic in some cases, it was useful to explore because it was illustrative of the competing processes that explain accumulation dependence in the model. Moreover,
another reason to understand this limiting case is that this is the scenario Arthern et al. (2010) proposed when providing a physical justification for Herron and Langway's (1980) low-density region model, which describes compaction rate as linearly related to accumulation rate (Eq. 1). As noted by Buizert et al. (2015), this equation, combined with density advection (which is also linearly proportional to accumulation rate) leads to accumulation having no impact on steady-state densities in this low-density regime. This effect manifests in our model as the two instances of accumulation rate, $\beta$, in Eq. 27 cancelling when the
surface grain size ($r_s^2$) is zero. Our results serve to highlight how, as first examined by Arthern et al. (2010) (their Appendix B), constitutive relations which describe firn compaction as linearly dependent on accumulation rate (e.g., Eq. 1) belie the crucial role played by grain size. In particular, usage of such constitutive relations implicitly assumes a negligibly small surface grain size, steady-state conditions, normal grain growth, and a particular form for the dependence of compaction on grain size (which we discuss more below).





An implication of these generally unrecognized assumptions underlying some widely used firn models is that models that include viscous firn compaction and grain size evolution (e.g., Arthern et al., 2010) are potentially capable of a much richer array of responses to accumulation rate than is usually recognized, if these assumptions were to be relaxed. For example, to correct for mismatch between density profiles observed in Antarctica and modelled by a reduced version of Arthern et al.'s (2010) full model, Ligtenberg et al. (2011) multiplied modelled compaction rates by a linear, empirically derived function of accumulation rate. Medley et al. (2020) improved upon this approach by instead tuning the original model's parameters to reduce mismatch between modelled and observed densities. Our work suggests that future work could apply similar approaches to a more complete model that relaxes the assumption of zero surface grain size, to examine if this reduces data-model mismatch.

While our model relaxes some important assumptions, others remain. These include the assumption that firn deforms viscously, air pressure is negligible, no water is present, rheological parameters are uniform and constant, firn grains grow via normal growth growth (with a growth exponent of 2), and firn viscosity is proportional to grain size.

We followed most previous firn models and assumed a viscous firn rheology (e.g., Stevens et al., 2020). An recent alternative approach instead assumes a plastic rheology and simulates the effect of air pressure on firn deformation in near-surface firn (Meyer et al., 2020). How our findings apply to such a model is yet to be determined.

Assuming dry firn compaction restricts the applicability of our results to regions where no significant melting takes place. In wet-snow zones the grain-scale processes that control compaction and grain growth will differ significantly from those in dry snow. Moreover, refreezing of meltwater contributes to densification. Understanding how grain growth, compaction and advection interact to control accumulation dependence in wet-snow zones is beyond our scope, but will likely become increasingly important as these regions grow in the future (e.g., Kittel et al., 2021; Gilbert and Kittel, 2021). Incorporating grain-size evolution into a model that accounts for meltwater percolation and refreezing (e.g., Meyer and Hewitt, 2017) may be an important step towards this.

For simplicity, and unlike firn compaction models that aim to accurately simulate porosity profiles, we used a uniform compaction coefficient, $k_c$, and a uniform stress exponent, $n$. Starting with Herron and Langway (1980), most firn models consider at least two porosity-defined regions with different compaction coefficients motivated by the different compaction mechanisms that operate in each region. Ignoring this complication does not qualitatively effect our key results relating to the accumulation sensitivity of firn thickness, but would need to be reconsidered when quantitatively comparing model output to observations. A related issue, which manifests when surface grain size is non-zero, is that the modelled vertical gradients of porosity and compaction velocity approach zero at the surface (Figure 2). This is counter to observations (e.g., Montgomery et al., 2018; Case and Kingslake, 2021), and is due to the compaction rate, $D\phi/Dt$, being zero at the surface due to a zero overburden stress (Eq. 2). Any firn compaction model that describes compaction as a function of overburden stress has the potential to suffer this limitation. Arthern and Wingham (1998) circumvented it using a constant high vertical strain rate in the near surface to account for fast compaction processes that cannot be described viscously, while Arthern et al. (2010) assumed zero grain size at the surface. Except in cases where we prescribe $r_s^2 = 0$ to explore accumulation dependence, we take neither approach here, but note that describing the variable compaction mechanisms that operate across different porosity ranges is





an important next step in understanding the accumulation dependence of firn thickness; in particular, our work highlights how
quantifying the grain-size dependence of compaction will be crucial for such efforts.

We also assumed that firn compaction is inversely proportional to the square of a characteristic grain size. Complications to this simple description could arise from non-uniform grain sizes (i.e which are inadequately described by a single-valued grain size variable), or from other compaction mechanisms that do not obey this simple inverse relationship.

We also assumed normal grain growth with an exponent of two (Eq. 8), as is appropriate for bubble-free ice with grain
growth driven by grain boundary migration to reduce interfacial energy (which is related to grain curvature). Azuma et al. (2012) show that in more realistic ice containing air bubbles the exponent could be much higher. A different grain-growth exponent would affect our results, but not change the conclusion that grain growth plays a role in the dependence of firn thickness on accumulation.

We have not explored the complications of multiple compaction regimes, different dependencies of compaction on grain
size, and different grain growth exponents or parameterizations. However, our work highlights the importance of doing so because commonly used constitutive relationships inspired by Herron and Langway (1980) make implicit assumptions related to these components of the system.

## 5   Conclusions and future work

The thickness of the firn layer in cold, dry accumulation zones, is controlled by a competition between downward advection
of firn and the compaction of each parcel of firn as it advects. To better understand the controls on advection and compaction, we analyzed a simplified model that is closely related to previous models (Arthern et al., 2010; Case and Kingslake, 2021). We scaled the model, solved model equations numerically, and derived and analyzed several simplified steady-state versions of the model. We draw two main conclusions: (1) the strength of the positive relationship between firn thickness and accumulation rate increases with the surface grain size, and (2) assumptions about grain size underlie some widely used compaction models
based on Herron and Langway (1980).

Future work could extend the model to include additional physics and apply the model to different scenarios. Model extensions could include employing a dynamically evolving temperature and varying rheological parameters between porosity-defined regions of the firn, although we anticipate that neither addition will affect our conclusions qualitatively. Additional simulations could explore model response to temporal changes in accumulation rate and temperature. Because the model in-
corporates accumulation differently than models that have been used for this purpose before (e.g., Zwally and Jun, 2002), comparison to those previous results could shed further light on the likely future response of firn to increases in accumulation, in particular how transients in grain size affect the temporal response of the ice thickness. As discussed above, another possible future use for this model, or derivatives of it, is to examine how relaxing the assumption of zero surface grain size affects the tuning of firn model parameters to observations of firn thickness (e.g., Ligtenberg et al., 2011; Medley et al., 2020).
The fact that modelled firn thickness depends on grain size at the surface has potentially significant implications because surface grain size varies in time and space due to meteorological conditions. Ongoing and future work by this team to test this



idea further include measuring deformation of firn with known grain size using phase-sensitive ice-penetrating radar (Case and Kingslake, 2021) co-located with grain size measurements from ice cores and conducting laboratory experiments compacting artificial firn samples with controlled grain sizes. Complimentary is analysis of recent compilations of firn thickness measure-
ments (e.g., Montgomery et al., 2018), in conjunction with modelled and measured accumulation rates, surface temperatures, and surface grain sizes.

## Appendix A: Derivation of the kinematic surface boundary condition

Here we use global mass conservation to derive Eq. 10, a kinematic condition for the rate of change in the length of the model domain, $h(t)$. Conservation of ice mass in the domain demands

$$\frac{\partial}{\partial t}\left[\int_0^{z_b}(1-\phi)\,dz\right]=0. \tag{A1}$$

This expression can be expanded using the Leibniz integration rule to give

$$(1-\phi_b)\dot{h}-\int_0^{z_b}\frac{\partial\phi}{\partial t}\,dz=0, \tag{A2}$$

where we have used $\dot{h}=\dot{z}_b$ in the first term. The second term can be found by substituting the definition of the material derivative into Eq. 6, rearranging, and recognizing that

$$\frac{\partial\phi}{\partial t}=(1-\phi)\frac{\partial w}{\partial z}-w\frac{\partial\phi}{\partial z}=\frac{\partial}{\partial z}[(1-\phi)w]. \tag{A3}$$

Substituting this into Eq. A2 and evaluating the integral gives

$$\int_0^{z_b}\frac{\partial\phi}{\partial t}\,dz=\int_0^{z_b}\frac{\partial}{\partial z}[(1-\phi)w]\,dz=(1-\phi_b)w(z_b)-(1-\phi_s)w(0), \tag{A4}$$

where $w(z_b)$ and $w(0)$ are the velocities at the bottom and top of the domain, respectively. The boundary condition on $w$ at the upper surface is $w(0)=b/(1-\phi_s)$ (Eq. 7). Substituting this into Eq. A4 and the result into Eq. A2 gives

$$(1-\phi_b)\dot{h}-(1-\phi_b)w(z_b)+b=0. \tag{A5}$$

Rearranging yields Eq. 10:

$$\dot{h}=w(z_b)-\frac{b}{1-\phi_b}\ . \tag{A6}$$

## Appendix B: Change of coordinates and numerical method

To take account of the temporally evolving domain length, we employ a change of vertical coordinate.





### B1 Partial derivatives

We normalize the vertical (nondimensional) coordinate, $z$, by $h(t)$, the nondimensional domain length. So that

$$z = h(t)\hat{z} , \tag{B1}$$

and $\hat{z} = 1$ and $\hat{z} = 0$ correspond to the lower and upper boundaries of the column, respectively. We then recast all model equations in term of this new vertical coordinate, $\hat{z}$. The time coordinate remains unchanged, but we write $\hat{t} = t$ for clarity. In what follows applying the multi-variable chain rule yields expressions for the partial directives with respect to $z$ and $t$ as functions of the scaled variables $\hat{z}$ and $\hat{t}$ and the partial derivatives with respect to $\hat{z}$ and $\hat{t}$. Applying the chain rule to expand the spatial derivative gives

$$\frac{\partial}{\partial z} = \frac{\partial \hat{z}}{\partial z}\frac{\partial}{\partial \hat{z}} + \frac{\partial \hat{t}}{\partial z}\frac{\partial}{\partial \hat{t}} . \tag{B2}$$

Therefore, given $\partial \hat{t}/\partial \hat{z} = 0$ and, from Eq. B1, $\partial \hat{z}/\partial z = 1/h$,

$$\frac{\partial}{\partial z} = \frac{1}{h}\frac{\partial}{\partial \hat{z}} . \tag{B3}$$

Applying the chain rule to expand the time derivative gives

$$\frac{\partial}{\partial t} = \frac{\partial \hat{z}}{\partial t}\frac{\partial}{\partial \hat{z}} + \frac{\partial \hat{t}}{\partial t}\frac{\partial}{\partial \hat{t}} . \tag{B4}$$

As $\partial \hat{t}/\partial t = 1$ and, from Eq. B1,

$$\frac{\partial \hat{z}}{\partial t} = -\frac{z\dot{h}}{h^2} = -\frac{\hat{z}\dot{h}}{h} , \tag{B5}$$

Eq. B4 shows that

$$\frac{\partial}{\partial t} = -\frac{\hat{z}\dot{h}}{h}\frac{\partial}{\partial \hat{z}} + \frac{\partial}{\partial \hat{t}} . \tag{B6}$$

### B2 Scaled Equations

The model equations are modified to account for the change of coordinates by substituting Equations B3 and B6 into the model equations (Equations 12, 14, 15, 17, 18 and 19).

The porosity equation (Eq. 12) becomes

$$\frac{\partial \phi}{\partial t} = \frac{1}{h}\frac{\partial}{\partial \hat{z}}\left[(1-\phi)w\right] + \left(\frac{\dot{h}\hat{z}}{h}\right)\frac{\partial \phi^*}{\partial \hat{z}} , \tag{B7}$$

where we also used Eq. 15 to simplify the expression. The stress equation (Eq. 14) becomes

$$\frac{\partial \sigma}{\partial \hat{z}} = -h^*(1-\phi^*) . \tag{B8}$$





The equation for the velocity gradient (Eq. 15) becomes

$$\frac{\partial w}{\partial \hat{z}} = -\frac{h}{\alpha} \frac{|\sigma|^n \phi^m}{r^2} \ . \tag{B9}$$

The grain size equation (Eq. 17) becomes

$$\frac{\partial r^2}{\partial t} = \left( \frac{\dot{h}\hat{z} - w}{h} \right) \frac{\partial r^2}{\partial \hat{z}} + (1 - \delta r^2) \ . \tag{B10}$$

The age equation (Eq. 18) becomes

$$\frac{\partial A}{\partial t} = 1 + \left( \frac{\dot{h}\hat{z} - w}{h} \right) \frac{\partial A}{\partial \hat{z}} \ . \tag{B11}$$

And finally, the domain thickness equation (Eq. 19) remains

$$\frac{\partial h}{\partial t} = w(z_b) - \frac{\beta}{1 - \phi_b} \ . \tag{B12}$$

### B3   Solution method

These six equations are solved with the method of lines to simulate how the six variables evolve in time and space during simulations. Specifically, the spatial domain is discretized into $N - 1$ grid spaces, connecting $N$ nodes. The four equations above containing time derivatives (Equations B7, B10, B11, and B12) are treated as $3N + 1$ coupled ODEs ($3N$ come from the $\phi$, $r^2$, and $A$ equations and one comes from the $h$ equation) using upwind finite difference (Kerschbaum, 2020). The coupled equations are solved simultaneously using the MATLAB ODE solver, ode15s. The remaining two equations (Equations B8 and B9) provide $\sigma$ and $w$ values used to compute the time derivatives. Spatial

To facilitate comparison between the results from simulations with different domain heights, all depths are converted from $\hat{z}$ back to $z$ with Eq. B1 before plotting.

### Appendix C: Accumulation independence of the full model when $r_s^2 = 0$

In Section 3.3 numerical solutions of the full model and inspection of a simplified ODE model (Eq. 27) indicates that firn thickness is independent of accumulation rate when $r_s^2 = 0$. For completeness, here we show the same result, starting from the full model and not making the simplifying assumptions about the velocity used to derive Eq. 27. This will also demonstrate that this finding is independent of the stress exponent $n$.

Starting with Eq. 12 and assuming a steady state gives Eq. 22a:

$$w \frac{d\phi}{dz} = -\frac{|\sigma|^n \phi^m (1 - \phi)}{\alpha r^2} \ . \tag{C1}$$

In what follows we derive expressions for $w$, $\sigma$, and $r^2$ in turn, then substitute them into the expression above to show that $\beta$ disappears when $r_s^2 = 0$. This is for the same reasons described in the main text – advection of porosity and of grain size balance each other.





Combining Equations 12 and 15 yields

$$\frac{\partial \phi}{\partial t} = \frac{\partial}{\partial z}\left[(1-\phi)w\right].$$ (C2)

Assuming a steady state and integrating vertically gives

$$(1-\phi)w = C_1,$$ (C3)

where $C_1$ is a constant of integration. Given the upper-surface boundary condition on the velocity ($w(0) = \beta/(1-\phi(z_b))$; Eq. 15.2), $C_1 = \beta$, yielding

$$w = \frac{\beta}{1-\phi}.$$ (C4)

Integrating Eq. 14 gives

$$\sigma = -\int_0^z (1-\phi)\,dz.$$ (C5)

Assuming a steady state in Eq. 17, and for simplicity assuming $\delta = 0$, yields

$$\frac{dr^2}{dz} = \frac{1}{w},$$ (C6)

which combined with Eq. C4 gives

$$\frac{dr^2}{dz} = \frac{1-\phi}{\beta}.$$ (C7)

Integrating and rearranging this expression gives

$$r^2 = r_s^2 + \frac{1}{\beta}\int_0^z (1-\phi)\,dz.$$ (C8)

Substituting Equations C4, C5, and C8 into Eq. C1, and assuming $r_s^2 = 0$ leaves

$$\frac{\beta}{1-\phi}\frac{d\phi}{dz} = -\beta\left[\int_z^{z_s} (1-\phi)\,dz\right]^{n-1}\frac{\phi^m(1-\phi)}{\alpha}.$$ (C9)

The accumulation rate $\beta$ appears on both sides of this expression – on the right due to grain size advection (Eq. C8) and on the left due to porosity advection (Eq. C4). Therefore, $\beta$ cancels and what remains is a differential equation for $\phi$ that is
independent of the accumulation rate. This is consistent with the simpler ODE model (Eq. 27) and with the numerical solutions of the full model showing a reduction in sensitivity as $r_s^2$ decreases (e.g., Figure 4a). Furthermore, Eq. C9 indicates that the fact that $\phi$ does not depend on $\beta$ in a steady state when $r_s^2 = 0$ is independent of the stress exponent $n$. This is consistent with the numerical results shown in Figure 7 (Section 3.4).





*Code availability.* All the code required to plot run the model and plot the figures in this manuscript can be found here:

https://doi.org/10.5281/zenodo.5213880

*Author contributions.* JK and EC initiated the study, CM advised on model physics, RS and JK developed the model and the code, JK led the modelling and writing, all authors contributed ideas and discussion and helped write the manuscript.

*Competing interests.* The authors declare no competing interests.

*Acknowledgements.* The authors acknowledge financial support from the National Science Foundation's Office of Polar Programs (grant
numbers OPP 19-35438 and PSU 5861-CU-NSF-8934).



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
