# Peer review of "Grain-size evolution controls the accumulation dependence of modeled firn thickness"

_The Cryosphere, 2022_

## Referee Comment (RC1)

**Review: "Grain-size evolution controls the accumulation dependence of modeled firn thickness"**

by Kingslake et al.

Submitted to *The Cryosphere*

**1 General**

In this paper, the authors analyze the role of snow accumulation rate on the compaction of dry firn layers on top of ice sheets. The authors' contribution is timely and important; as summarized in their introduction "Some [models] treat accumulation as a boundary condition, as it is in other ice-deformation modelling contexts. Others include $b$ [accumulation rate] in their constitutive relations." (with the citations removed and two parenthetical additions). This is important because in Herron and Langway (1980), the gold standard for firn compaction, the empirical compaction constitutive relation is proportional to accumulation rate, meaning that if it stops snowing, compaction ceases. However, in the physical system, compaction should continue if snow accumulation stops; this is the problem that Kingslake et al. address in this paper. Building from Appendix B of Arthern et al. (2010), the authors develop a full numerical simulation of very sensible compaction equations including grain growth and complement these results with steady state ordinary differential equation (ODE) solutions to reduced models. Arthern et al. (2010) contends that if the grain size ($r_s^2$ in the parlance) is near zero at the surface, then the system approaches Herron and Langway (1980), with a direct dependence on the accumulation rate. Here Kingslake et al. show that in the $r_s^2 \to 0$ limit, the effects of porosity advection and grain-size advection cancel and the model firn depth is independent of the accumulation rate — another puzzling result of the Arthern et al. (2010) work. Kingslake summarizes "models that include viscous firn compaction and grain size evolution (e.g. Arthern et al. (2010)) are potentially capable of a much richer array of response to accumulation rate than is usually recognized." This is a very nice paper that cogently examines the formulation of the predominant firn compaction model. With some small tweaks, this paper is certainly worthy of publication in *The Cryosphere*.

**2 Remarks**

1. The model observation that the firn depth $z_{830}$ increases with accumulation rate is interesting and challenges my intuition. The weakest dependence comes from Arthern et al. (2010) with $r_s^2 \to 0$, where the depth is independent of $b$. Yet many of the places with the deepest firn columns, e.g. domes of East Antarctica, have very low accumulation rates, leading me to think that there is a regime where $z_{830} \sim 1/b$. Is that misguided? This would seem to imply that if the accumulation rate increased in East Antarctica, then the firn depths would also increase. This seems to be backwards from the current firn thickness / accumulation rates in Antarctica, where thinner firn thicknesses exist on the coasts where it snows more.

2. Interestingly, rather than leaving the temperature constant for the different accumulation rates, am I understanding correctly that the temperature also changes? It seems like it might be better to change one thing at a time to see the effect of that change more clearly? Or do you change them independently? More generally, I am curious about the effects of temperature on the results.

**3 Specific comments**

1. Figure 2(left): the ODE and full compaction model results are indeed indistinguishable. Maybe make one dashed (narrow) and the other solid (wide) and embed them? Or use symbols for the full model?

2. Figure 2(right): would the absolute difference normalized by the absolute maximum value of the quantity be more useful?

3. Figures 3, 4: it could be useful to mark the location $z_{830}$ on the plots using a symbol, so that the trend from the inset is visually clear. Maybe it could go on the velocity side, so that it doesn't get compressed at the bottom of the porosity plot.

**References**

R. J. Arthern, D. G. Vaughan, A. M. Rankin, R. Mulvaney, and E. R. Thomas. In situ measurements of Antarctic snow compaction compared with predictions of models. *J. Geophys. Res.*, 115(F3), 2010. doi: 10.1029/2009JF001306.

M. M. Herron and C. C. Langway. Firn densification: an empirical model. *J. Glaciol.*, 25(93):373–385, 1980. doi: 10.1017/S0022143000015239.

---

## Referee Comment (RC2)

Review of: "Grain-size evolution controls the accumulation dependence of modeled firn thickness"
by Jonathan Kingslake, Robert Skarbek, Elizabeth Case, and Christine McCarthy

**Summary**

This paper examines how the accumulation rate, which is a commonly used parameter in firn-densification models, influences the firn thickness. The authors develop a non-dimensional firn model that allows them to consider how the advection rate affects the firn. They find that the accumulation rate controls the downward advection of porosity and grain size, and higher accumulation rates cause faster downward advection of firn. This faster advection of high-porosity firn from the surface leads to greater firn thickness. Conversely, grain growth leads to larger grains, which slows firn compaction; faster advection leads to smaller grains (faster compaction) through the depth of the firn column and thereby a shallower firn column. When the surface grain size is zero, these effects balance exactly.

In general, I found this paper to be insightful and well written. Its scientific arguments were convincing. I appreciate the method used (a non-dimensionalized model) in that it clearly elucidates how inherent assumptions in firn models manifest in model outputs. The authors' detailed analyses of these implicit assumptions are an important contribution to the field. This paper will help push firn modeling towards more physically based models that describe microstructural processes. I am happy to recommend that it be published with minor revisions.

**General Comments**

- The model used assumes that the firn is isothermal. The conclusion states that including heat transfer should not qualitatively affect the results, but does not include any specifics as to why I should believe that. For example, the grain growth rate is determined by an Arrhenius dependence, which means that grain growth rates vary significantly through the year. I think it will be not too much work or new text to strengthen the argument that the isothermal assumption is valid here and discuss possible implications. (This comment was specifically spurred by the discussion at Line 172-173 – after reading that I was expecting more discussion about temperature effects.)

- I would appreciate a bit of discussion of what you mean (or has been meant historically in the studies you cite) by 'grain size'. I hear some people in the firn community talk broadly about 'grain size' and others talk about specific, measurable microstructural properties of the firn (e.g. specific surface area, optical-equivalent grain size, etc.). When we talk about normal, Gow-type grain growth, what is the property we are actually referring to? I think this is important given that a take-away message of your paper is that models need to be considering grain size. Can you provide recommendations of what field studies should be measuring?

- You provide a single value for $k_c$, which is the value suggested by Arthern et al. (2010) for low density snow (i.e. stage 1 densification). The value of $k_c$ for higher density firn ($3.7*10^9$) is significantly lower. Can you justify your decision to use the single, low-density-snow, value for the entire firn column? I would suspect that this would consistently lead to modeled values of

$z_{830}$ that are too shallow. Will this affect your results, especially concerning the point at which the grain-size- and porosity-advection balance exactly? (Ok, now I see that it is mentioned near the end of the paper, but I would still like to see a bit more discussion of this – using a different $k_c$ for higher density firn will change the vertical velocity for a substantial portion of the firn column)

- Figures: I found that the font size on figure axis labels and ticks was too small (figures 1,2, 5, 6, 7); I had too zoom in on the pdf to read them. (Figure 8 is good!)

**Specific Comments**
Abstract: you state, "the downward advection of porosity and of grain size are both affected by b, but have opposing impacts on firn thickness. The net result is that firn thickness increases with b and that the strength of this dependence increases with the surface grain size." I don't disagree, and it is clear to me what you mean after I have read the paper. But, I think this could be clarified here by explicitly stating how the porosity and grain size affect firn thickness (i.e., larger grains = slower compaction). When you say increases with surface grain size, do you mean increases with increasing surface grain size?

Line 42: should be "and/or" grain size

L52: You could mention that some models circumvent this issue by using the average accumulation over the lifetime of the firn layer, and is done in Li and Zwally and Stevens et al.

L57: It is not clear to me what you mean by your statement that a Lagrangian approach obscures advection. I agree that the advection is more explicit in an Eulerian model, but the advection rate and vertical velocity of layers can easily be tracked in a Lagrangian framework as well.

L177: sentence starting with 'While' – a bit of a run on sentence, can you break up to make it easier to parse for the reader? (I had to read it several times to get the gist.)

L182: do you mean that the firn compaction rate will approach zero? And can you be more specific about what 'at depth' means? i.e. at the bottom of the model domain, or beyond some defined threshold?

L186: do you mean grain growth rate increases? "Grain growth increases" implies that there is more grain growth, which I am not sure how to interpret – more of the grains are experiencing growth?

L191: "combination of conditions are likely" → 'is likely'

L226: Is the fact that that nondimensional time is 0.83 purely coincidental or related to the fact that you are nondimensionalizing and 0.83 is the close off porosity? (Or, perhaps I am asking if you prescribe a densification rate change at 830 kg/m$^3$.)

L226/Figure 2: I am a surprised to see that the porosity becomes zero in the lower region – given the prescribed conditions and a domain depth of 100m, I would expect that there is a small bit of porosity remaining at 100m (i.e. that the firn would not have reached 918 kg/m$^3$ at that depth) – is this a result of the use of the low-density-snow coefficient I mentioned above?

Line 233/Figure 2: Is there anything to be read from the structure of the mismatch (e.g., porosity at z=0.3?)

Figure 3: can you explain the inset more clearly? I think that you mean the three dotted lines the same in each inset – is that correct?

Figure 5 is a bit challenging to interpret because it is plotting "dependence" – I understand how you calculated it, but I suggest adding to the caption to include the takeaway message from that figure, i.e. I think that the figure is showing that as surface grain size increases, the firn thickness is more dependent on the accumulation rate (?). Perhaps describe in the context of Figure 6a – a horizontal line drawn across at $r^2_s$ = 0.01 will not change the value of $z_{830}$ very much, while a line drawn across $r^2_s$ = 0.09 will lead to increasing $z_{830}$ with increasing B. (Hopefully I am interpreting that correctly.)

L403: I think this no-accumulation dependence is also demonstrated in your Figure 1?

L434: 'effect' → affect

L437: This observation that compaction rate is zero at the surface is astute. I am not sure where/how, but potentially it deserves to be highlighted a bit more.

L538: This paragraph appears to be unfinished.

---

## Author Response (AR1)

**Response to reviewers for "Grain-size evolution controls the accumulation dependence of modeled firn thickness" (tc-2022-13)**

Our responses are shown in **bold**.

**Reviewer 1**

**1 General**

In this paper, the authors analyze the role of snow accumulation rate on the compaction of dry firn layers on top of ice sheets. The authors' contribution is timely and important; as summarized in their introduction "Some [models] treat accumulation as a boundary condition, as it is in other ice-deformation modelling contexts. Others include b [accumulation rate] in their constitutive relations." (with the citations removed and two parenthetical additions). This is important because in Herron and Langway (1980), the gold standard for firn compaction, the empirical compaction constitutive relation is proportional to accumulation rate, meaning that if it stops snowing, compaction ceases. However, in the physical system, compaction should continue if snow accumulation stops; this is the problem that Kingslake et al. address in this paper. Building from Appendix B of Arthern et al. (2010), the authors develop a full numerical simulation of very sensible compaction equations including grain growth and complement these results with steady state ordinary differential equation (ODE) solutions to reduced models. Arthern et al. (2010) contends that if the grain size (rs2 in the parlance) is near zero at the surface, then the system approaches Herron and Langway (1980), with a direct dependence on the accumulation rate. Here Kingslake et al. show that in the rs2 → 0 limit, the effects of porosity advection and grain-size advection cancel and the model firn depth is independent of the accumulation rate — another puzzling result of the Arthern et al. (2010) work. Kingslake summarizes "models that include viscous firn compaction and grain size evolution (e.g. Arthern et al. (2010)) are potentially capable of a much richer array of response to accumulation rate than is usually recognized." This is a very nice paper that cogently examines the formulation of the predominant firn compaction model. With some small tweaks, this paper is certainly worthy of publication in The Cryosphere.

**Thank you for the review and the positive words about the paper. We are pleased to hear you find it timely and important.**

**2 Remarks**

1. The model observation that the firn depth $z_{830}$ increases with accumulation rate is interesting and challenges my intuition. The weakest dependence comes from Arthern et al. (2010) with $r_{s2} \rightarrow 0$, where the depth is independent of b. Yet many of the places with the deepest firn columns, e.g. domes of East Antarctica, have very low accumulation rates, leading me to think that there is a regime where $z_{830}$ ~ 1/b. Is that misguided? This would seem to imply that if the accumulation rate increased in East Antarctica, then the firn depths would also increase. This seems to be backwards from the current firn thickness / accumulation rates in Antarctica, where thinner firn thicknesses exist on the coasts where it snows more.

**Yes, this model predicts that firn thickness increases with increasing accumulation rate. The primary reason for the thicker firn in interior East Antarctica versus other parts of the ice sheets is the lower surface temperatures. Firn compaction rate is a strong function of temperature, so the low surface temperatures (and hence low temperatures throughout the firn column) lead to slow firn compaction. To isolate the relationship between accumulation rate and firn thickness, we left temperature uniform and constant throughout the modeling. In observational records of firn thickness the relationship between accumulation rate and firn thickness is made less clear by the positive relationship between temperature and accumulation rate - in general, higher temperature leads to higher accumulation. So in cold places like East Antarctica the low temperatures cause slow compaction but they also cause low accumulation rate, the effect of which in this case is insufficient to counteract the effect of the low temperatures and the net result is relatively thick firn.**

**This is discussed very briefly in the new paragraph quoted below in lines 431-440.**

2. Interestingly, rather than leaving the temperature constant for the different accumulation rates, am I understanding correctly that the temperature also changes? It seems like it might be better to change one thing at a time to see the effect of that change more clearly? Or do you change them independently? More generally, I am curious about the effects of temperature on the results.

**To isolate the relationship between accumulation rate and firn thickness, we assumed a uniform and constant temperature throughout. We leave exploration of temperature effects to future work.**

**We have added the following paragraph to the discussion (L431-440)section to discuss this issue.**

**"To isolate the effects of accumulation rate on firn thickness we assumed a uniform and constant temperature. However, temperature is a first-order control on firn thickness in this model, through its impact on grain growth and on firn compaction (Section 2.3). Surface temperatures vary regionally with climate. This variability would need to be taken into account in any attempt to compare model results to observations (e.g., Montgomery et al., 2018), particularly because accumulation rates generally increase with increasing temperature (e.g., Frieler et al., 2015, Dalaiden et al., 2020), complicating the simple relationship between accumulation and firn thickness predicted by our model. Temperature also varies in time, causing transient vertical variability in temperature throughout the firn column, in part, through advection of heat. Further model development and analysis will be required to assess how the modelled accumulation dependence of firn thickness differs in this scenario, in particular in the case when accumulation also varies in time. The latter, presents the possibility of complex interplay between advection of porosity, grain size, and heat. We leave exploration of this to future work."**

**We hope that the new paragraph, along with the mention of this isothermal assumption in the second paragraph in section 2.1 (L80-81) make it clear to the reader that we are making this assumption.**

**Specific comments**

Figure 2(left): the ODE and full compaction model results are indeed indistinguishable. Maybe make one dashed (narrow) and the other solid (wide) and embed them? Or use symbols for the full model?

**We were unable to find a way of making the two sets of curves distinguishable: using markers did not help because there are many grid points and therefore markers merge together to make one thick line, while using two solid, very thin lines for each curve detracts from the readability of the plot, as currently the line thicknesses and styles uniquely distinguishes the curves without the need for the colors, helping color-blind readers. If we were to remove the line styles this benefit would be lost.**

**We decided to not change the line styles, but are happy to revisit this decision if the reviewer or the editor would like us too, and have suggestions for the best compromise of readability and clarity.**

Figure 2(right): would the absolute difference normalized by the absolute maximum value of the quantity be more useful?

**Normalizing this plot would scale the curves, but not change the plot significantly because the model is nondimensionalized. We chose not to make this change because the original approach seems like the simpler, and clearer, of the two options.**

Figures 3, 4: it could be useful to mark the location z830 on the plots using a symbol, so that the trend from the inset is visually clear. Maybe it could go on the velocity side, so that it doesn't get compressed at the bottom of the porosity plot.

**This is a good suggestion. We have added markers to the right hand panels and a horizontal line to the left panels, and updated the captions appropriately.**

**References**

R. J. Arthern, D. G. Vaughan, A. M. Rankin, R. Mulvaney, and E. R. Thomas. In situ measurements of Antarctic snow compaction compared with predictions of models. J. Geophys. Res., 115(F3), 2010. doi: 10.1029/2009JF001306.

M. M. Herron and C. C. Langway. Firn densification: an empirical model. J. Glaciol., 25(93):373–385, 1980. doi: 10.1017/S0022143000015239.

**Reviewer 2**

**Summary**

This paper examines how the accumulation rate, which is a commonly used parameter in firn-densification models, influences the firn thickness. The authors develop a non-dimensional firn model that allows them to consider how the advection rate affects the firn. They find that the accumulation rate controls the downward advection of porosity and grain size, and higher accumulation rates cause faster downward advection of firn. This faster advection of high-porosity firn from the surface leads to greater firn thickness. Conversely, grain growth leads to larger grains, which slows firn compaction; faster advection leads to smaller grains (faster compaction) through the depth of the firn column and thereby a shallower firn column. When the surface grain size is zero, these effects balance exactly.

In general, I found this paper to be insightful and well written. Its scientific arguments were convincing. I appreciate the method used (a non-dimensionalized model) in that it clearly elucidates how inherent assumptions in firn models manifest in model outputs. The authors'

detailed analyses of these implicit assumptions are an important contribution to the field. This paper will help push firn modeling towards more physically based models that describe microstructural processes. I am happy to recommend that it be published with minor revisions.

**General Comments**

- The model used assumes that the firn is isothermal. The conclusion states that including heat transfer should not qualitatively affect the results, but does not include any specifics as to why I should believe that. For example, the grain growth rate is determined by an Arrhenius dependence, which means that grain growth rates vary significantly through the year. I think it will be not too much work or new text to strengthen the argument that the isothermal assumption is valid here and discuss possible implications. (This comment was specifically spurred by the discussion at Line 172-173 – after reading that I was expecting more discussion about temperature effects.

**We agree that this suggestion that temporally or spatially varying temperatures will not affect our conclusions requires more support. We have concluded that this would require more model development and analysis, so have chosen to save this for future work.**

**We removed the phrase in the conclusions stating that we do not expect addition of a dynamic temperature to affect our results.**

**We also added a new paragraph to the discussion section (L431-440) discussing the isothermal assumption:**

**"To isolate the effects of accumulation rate on firn thickness we assumed a uniform and constant temperature. However, temperature is a first-order control on firn thickness in this model, through its impact on grain growth and on firn compaction (Section 2.3). Surface temperatures vary regionally with climate. This variability would need to be taken into account in any attempt to compare model results to observations (e.g., Montgomery et al., 2018), particularly because accumulation rates generally increase with increasing temperature (e.g., Frieler et al., 2015, Dalaiden et al., 2020), complicating the simple relationship between accumulation and firn thickness predicted by our model. Temperature also varies in time, causing transient vertical variability in temperature throughout the firn column, in part, through advection of heat. Further model development and analysis will be required to assess how the modelled accumulation dependence of firn thickness differs in this scenario, in particular in the case when accumulation also varies in time. The latter, presents the possibility of complex interplay between advection of porosity, grain size, and heat. We leave exploration of this to future work."**

- I would appreciate a bit of discussion of what you mean (or has been meant historically in the studies you cite) by 'grain size'. I hear some people in the firn community talk broadly about

'grain size' and others talk about specific, measurable microstructural properties of the firn (e.g. specific surface area, optical-equivalent grain size, etc.). When we talk about normal, Gow-type grain growth, what is the property we are actually referring to? I think this is important given that a take-away message of your paper is that models need to be considering grain size. Can you provide recommendations of what field studies should be measuring?

**Normal grain growth is usually described in terms of the mean grain size. To clarify this we have added the following in the Methods section (L111-112) "This approach assumes that we can characterize grain size by the mean grain radius r, ignoring complications associated with more realistic grain-size distributions (Kipfstuhl et al., 2009)."**

**However, this does not address the difficulty of measuring the mean grain size. Attempting to mitigate the effect of thin sections tending to under-represent the width of grains, Gow (1969) computed mean grain size from the 50 largest grain in each section. However, Kipfstuhl et al. (2009) showed that this can introduce a sampling bias that obscures the role that recrystallization plays in reducing grain size, concluding that (in reference to Gow (1969)) *An arbitrarily selected population does not guarantee that the behavior of the deforming polycrystalline material is described well.*
Rather than discuss this debate in detail, we add the following to the discussion to acknowledge that alternatives to normal grain growth are possible and that replacing this component of our model would affect the detail of our results: (L472-476)**

**"Moreover, Kipfstuhl et al. (2009) show evidence of pervasive recrystallization in firn and highlight problems associated with the measurements of grain size used previously as evidence for normal grain growth in firn Gow (1969). They conclude that the assumption of normal grain growth in firn should be revisited. A different normal-grain-growth exponent or a different approach to modelling grain-size evolution would affect our results, but not change our broader conclusion that grain growth plays a significant role in the dependence of firn thickness on accumulation."**

**This adds weight to our argument that we need a better understanding of grain size evolution in firn if we are to build physically based models that can determine how firn thickness will change as accumulation rates change.**

- You provide a single value for kc, which is the value suggested by Arthern et al. (2010) for low density snow (i.e. stage 1 densification). The value of kc for higher density firn (3.7*109) is significantly lower. Can you justify your decision to use the single, low-density-snow, value for the entire firn column? I would suspect that this would consistently lead to modeled values of z830 that are too shallow. Will this affect your results, especially concerning the point at which the grain-size- and porosity-advection balance exactly? (Ok, now I see that it is mentioned near the end of the paper, but I would still like to see a bit more discussion of this – using a different kc for higher density firn will change the vertical velocity for a substantial portion of the firn column)

**Yes, using a different value of kc would quantitatively affect the results. For example, changing this value uniformly would scale the porosity profiles linearly and affect the firn thickness, and using a different value of kc at lower porosity (higher density) would affect the shape of the porosity profiles and firn thickness. However, neither change to kc would affect our overall conclusions about the dependence of firn thickness on accumulation, and how this dependence itself depends on surface grain size.**

**As this is already mentioned explicitly in the discussion (L454-456: "Ignoring this complication does not qualitatively affect our key results relating to the accumulation sensitivity of firn thickness, but would need to be reconsidered when quantitatively comparing model output to observations.") We have not added any more discussion on this topic to the paper.**

- Figures: I found that the font size on figure axis labels and ticks was too small (figures 1,2, 5, 6, 7); I had too zoom in on the pdf to read them. (Figure 8 is good!)

**These figures have been updated.**

**Specific Comments**

Abstract: you state, "the downward advection of porosity and of grain size are both affected by b, but have opposing impacts on firn thickness. The net result is that firn thickness increases with b and that the strength of this dependence increases with the surface grain size." I don't disagree, and it is clear to me what you mean after I have read the paper. But, I think this could be clarified here by explicitly stating how the porosity and grain size affect firn thickness (i.e., larger grains = slower compaction). When you say increases with surface grain size, do you mean increases with increasing surface grain size?

**Yes, we do. This has been edited to make this clearer: (L9-10) "the strength of this dependence increases with increasing surface grain size."**

Line 42: should be "and/or" grain size

**Corrected (L42)**

L52: You could mention that some models circumvent this issue by using the average accumulation over the lifetime of the firn layer, and is done in Li and Zwally and Stevens et al.

**We have added (L53-54): "Some models circumvent this issue by using the mean accumulation over the time since each firn layer was deposited"**

L57: It is not clear to me what you mean by your statement that a Lagrangian approach obscures advection. I agree that the advection is more explicit in an Eulerian model, but the advection rate and vertical velocity of layers can easily be tracked in a Lagrangian framework as well.

**Agreed, the advection of each layer is available for analysis in a numerical simulation of firn using a model that takes a Lagrangian framework. We intended to say something more specific: that a Lagrangian framework makes it difficult to analytically (i.e. without numerical methods) examine the model equations to compare terms describing advection with other terms. This was the more restricted sense in which we meant that such an approach obscures advection.**

**We have made the statement more specific to read (L59-61):**

**"...most take a Lagrangian approach (and so track each firn layer individually, preventing analytical examination of model equations to isolate the role of advection), makes unravelling the influence of..."**

L177: sentence starting with 'While' – a bit of a run on sentence, can you break up to make it easier to parse for the reader? (I had to read it several times to get the gist.)

**We split this sentence into two and rephrased to shorten (L184-186):**

**"Note that competition between the effects of grain-size evolution and advection manifests here in terms of model scales and nondimensional parameters. Later we will discuss the same competition in more detail when it reappears while considering the effect of changing nondimensional model inputs between simulations (specifically, $r_s^2$ and $\beta$)."**

L182: do you mean that the firn compaction rate will approach zero? And can you be more specific about what 'at depth' means? i.e. at the bottom of the model domain, or beyond some defined threshold?

**Yes, thanks, the word 'porosity' was missing from this sentence. And yes, we mean at the bottom of the model domain. The sentence now reads (L187-189):**

**"However, even in the low temperature climate, $\alpha < 1$, indicating that compaction is large compared to advection and that firn porosity will usually closely approach zero at the bottom of the model domain in our simulations."**

L186: do you mean grain growth rate increases? "Grain growth increases" implies that there is more grain growth, which I am not sure how to interpret – more of the grains are experiencing growth?

**Yes, thanks, we mean grain growth rate increases. This has been corrected (L193).**

L191: "combination of conditions are likely"à'is likely'

**Corrected (L198-199)**

L226: Is the fact that that nondimensional time is 0.83 purely coincidental or related to the fact that you are nondimensionalizing and 0.83 is the close off porosity? (Or, perhaps I am asking if you prescribe a densification rate change at 830 kg/m3.)

**This is just a coincidence. To avoid people interpreting this otherwise we have changed this to read (L234-235):**

**"...after a nondimensional time of around 0.8 (800 years)."**

L226/Figure 2: I am a surprised to see that the porosity becomes zero in the lower region – given the prescribed conditions and a domain depth of 100m, I would expect that there is a small bit of porosity remaining at 100m (i.e. that the firn would not have reached 918 kg/m3 at that depth) – is this a result of the use of the low-density-snow coefficient I mentioned above?

**Yes, this is due to our choice of model parameters. Our focus is on the dependence of firn thickness on accumulation and grain size and this is not affected by the choice of model parameters (e.g., k_c), as described above and mentioned in the discussion. To avoid distracting the reader from the main point of this section, we opted to not switch the focus of this part of the manuscript to the relatively low firn thickness, even though, as we discuss in the discussion, we would need to carefully choose the correct k_c (and all the other model parameters) if we were to compare model predictions to observations quantitatively.**

Line 233/Figure 2: Is there anything to be read from the structure of the mismatch (e.g., porosity at z=0.3?)

**The structure of the residuals between the ODE model solution and the approximately steady-state solution of the full model (Figure 2b) reflect the structure of the model. Three of the model variables - porosity, grain size and age - approach a steady state in the full model by approaching a situation where the advection of each quantity balances compaction, grain growth, and aging, respectively. This sets the spatial distribution of the residuals to first order; places where advection (and therefore, either compaction, grain growth, or aging, because these balance) are large, are the places where the residuals will be large.**

**Because advection is proportional to the gradient of the variable, the spatial distributions of the porosity, grain size and age residuals are approximately proportional to the gradient of the relevant variable.**

**Overburden pressure and velocity are different from the other five variables, in the sense that they do not evolve through a combination of advection and another process, they are computed from the other variables. Their residuals in Figure 2b reflect this: the overburden stress residuals are approximately proportional to the integral of the porosity residuals, reflecting the simple dependence of stress gradient on porosity (eqn 14a), while the velocity residuals are more complex, reflecting the dependence of velocity on grain size, porosity, and stress (eqn 15a).**

**Since these considerations do not provide insights beyond what is already evident in the model equations, we chose not to add any text discussing this to the manuscript.**

Figure 3: can you explain the inset more clearly? I think that you mean the three dotted lines the same in each inset – is that correct?

**We edited the last sentence of Figure 3's caption to read: "The three dotted curves are the same in each inset and the solid curve in each inset corresponds to the results shown in the respective row."**

Figure 5 is a bit challenging to interpret because it is plotting "dependence" – I understand how you calculated it, but I suggest adding to the caption to include the takeaway message from that figure, i.e. I think that the figure is showing that as surface grain size increases, the firn thickness is more dependent on the accumulation rate (?). Perhaps describe in the context of Figure 6a – a horizontal line drawn across at r2s = 0.01 will not change the value of z830 very much, while a line drawn across r2s = 0.09 will lead to increasing z830 with increasing B. (Hopefully I am interpreting that correctly.)

**Yes, this is what we are intending to show. Thanks for highlighting the chance to make this relationship between grain size, accumulation and firn thickness clearer and to make the link between Figures 5 and 6 clearer.**

**We added the following to the end of Figure 5's caption:**
**"Note that as surface grain size increases, the firn thickness is more dependent on the accumulation rate. This is the same relationship shown in Figure 6."**

**And we added the following to the end of figure 6's caption:**
**"All three panels show the same relationship between surface grain size, accumulation and firn thickness: as surface grain size increases, the firn thickness is more dependent on the accumulation rate (see also Figure 5)."**

L403: I think this no-accumulation dependence is also demonstrated in your Figure 1?

**Figure 1 does not demonstrate this as it only shows the results of simulations that used the same accumulation rate (but different grid spacings). No changes were made to the manuscript.**

L434: 'effect'àaffect

**Corrected (L454)**

L437: This observation that compaction rate is zero at the surface is astute. I am not sure where/how, but potentially it deserves to be highlighted a bit more.

**Thank you. This has been pointed out before, for example, in the Physics of Glaciers, Cuffey and Paterson (1994). We think this is the appropriate place in the paper for this discussion and are unsure if it needs to be reiterated elsewhere. Therefore we have not made any changes to the manuscript, except to add a citation to that textbook (L460).**

L538: This paragraph appears to be unfinished.

Corrected by deleting the sentence, as it was unneeded (L561).

---

## Referee Report (RR1)

Review of (revised): "Grain-size evolution controls the accumulation dependence of modeled firn thickness"

by Jonathan Kingslake, Robert Skarbek, Elizabeth Case, and Christine McCarthy

I thank the authors for their thoughtful responses to the points I raised in my previous review. They have adequately addressed the concerns and questions I had. I have no more comments, and I am happy to recommend this paper for publication.